# KEA: Keeping Exploration Alive by Proactively Coordinating Exploration Strategies in Novelty-based Exploration

## Abstract

In continuous control tasks, Soft Actor-Critic (SAC) has achieved notable success by balancing exploration and exploitation. However, SAC struggles in sparse reward environments, where infrequent rewards hinder efficient exploration. While novelty-based exploration methods help address this issue by encouraging the agent to explore novel states, they introduce challenges, such as the difficulty of setting an optimal reward scale and managing the interaction between novelty-based exploration and SAC's stochastic policy. These complexities often lead to inefficient exploration or premature convergence and make balancing exploration-exploitation challenging. In this paper, we propose KEA (Keeping Exploration Alive) to tackle the inefficiencies in balancing the exploration-exploitation trade-off when combining SAC with novelty-based methods. KEA introduces an additional co-behavior agent that works alongside SAC and a switching mechanism to facilitate proactive coordination between exploration strategies from the co-behavior agent and the SAC agent with novelty-based exploration. This coordination allows the agent to maintain stochasticity in high-novelty regions, preventing premature convergence and enhancing exploration efficiency. We first analyze the difficulty of balancing exploration-exploitation when combining SAC with novelty-based methods in a 2D grid environment. We then evaluate KEA on sparse reward control tasks from the DeepMind Control Suite and compare against two state-of-the-art novelty-based exploration baselines — Random Network Distillation (RND) and NovelD. KEA improves episodic rewards by up to 119% over RND and 28% over NovelD, significantly improving learning efficiency and robustness in sparse reward environments.

## 1 Introduction

Despite the success of deep reinforcement learning (RL) in continuous control tasks, such as robotic manipulation, these methods often rely on manually designed dense rewards (Zhou et al. (2023); Zhou & Held (2023); Zhang et al. (2023); Yang et al. (2024)), which require task-specific domain expertise. This reliance makes them impractical for real-world applications and difficult to generalize across diverse tasks. To reduce the reliance on handcrafted dense rewards, early works have focused on sparse reward settings, where rewards are rare or difficult to obtain. While this reduces the need for expert-designed dense rewards, it makes learning inefficient due to the lack of informative reward signals. In this setting, basic RL exploration methods, such as stochastic sampling (Tokic (2010); Bridle (1989)) and unstructured additive noise (Silver et al. (2014)), often fail because the agent struggles to identify beneficial actions with limited feedback. In this context, Soft Actor-Critic (SAC) (Haarnoja et al. (2018)) has demonstrated significant success in continuous control tasks (Zhou et al. (2023); Zhou & Held (2023); Yang et al. (2022)) by optimizing both exploration and exploitation through its stochastic policies. However, in sparse reward settings, even SAC struggles due to the lack of frequent reward signals, making the agent difficult to explore efficiently.

Previous works (Ng et al. (1999); Hu et al. (2020); Ladosz et al. (2022)) applied reward shaping to mitigate this inefficiency. However, shaped rewards can misalign the agent's objective from the true objective of the task (Irpan (2018); Popov et al. (2017)). Solving sparse reward tasks is crucial to ensure agents learn the correct objectives.

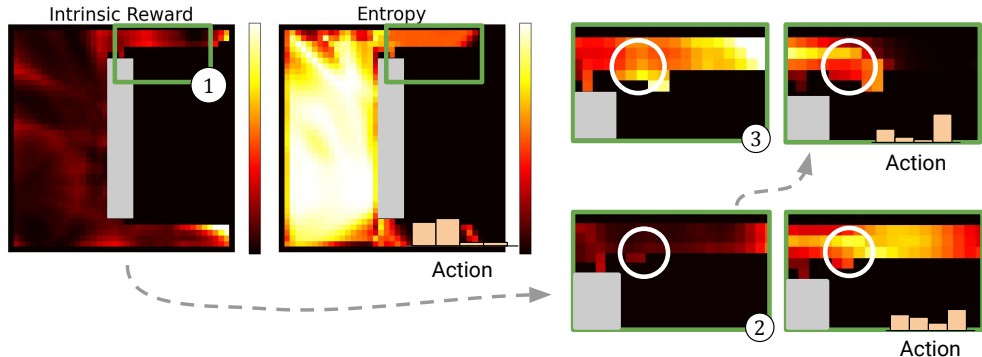

Figure 1: **Interactions between different exploration strategies**. Exploration behavior is influenced by two primary factors: novelty-based intrinsic rewards, which drive exploration toward novel states, and stochasticity in the policy, often maintained through entropy. Each histogram shows action probabilities for *move right*, *move left*, *move up*, and *move down* at different stages: (1) high intrinsic rewards with low entropy, (2) decreasing intrinsic rewards with increasing entropy, and (3) the discovery of an unvisited region.

One promising solution to the challenge of sparse rewards is to augment (extrinsic) rewards with intrinsic rewards, purposefully designed to encourage exploration. Curiosity-based methods (Pathak et al. (2017); Burda et al. (2018a)) leverage a learned dynamics model of the environment to predict future states, deriving intrinsic rewards from the prediction errors. Similarly, novelty-based methods (Burda et al. (2018b); Badia et al. (2020)) compute intrinsic rewards based on the novelty of visited states, encouraging the agent to explore unfamiliar regions. However, the agent may waste resources discovering novelty retroactively, which affects exploration efficiency because the agent cannot determine how novel an unvisited state is until it is explored. To address this inefficiency, NovelD (Zhang et al. (2021)), combines novelty differences with episodic counting-based bonuses to encourage the agent to explore the boundary between explored and unexplored regions, promoting more efficient exploration.

While novelty-based exploration methods have been shown to improve exploration when coupled with an on-policy RL method such as PPO (Schulman et al., 2017), applying these reward-based methods to sample-efficient modern off-policy learning algorithms is challenging for several reasons. Soft Actor-Critic (SAC) is well-documented for its sensitivity to reward scaling (Haarnoja et al. (2018)), and this sensitivity extends to intrinsic rewards. The non-stationary nature of intrinsic rewards creates a rapidly shifting objective, making it even more difficult to set an optimal reward scale. Inappropriate scaling can lead to excessive randomness when set too low, or premature convergence (to novel regions) when set too high, further complicating the exploration-exploitation trade-off.

Compounding these challenges, the interaction between novelty-based exploration and exploration via stochastic policy adds further complexity to the exploration behavior, as shown in Fig. 1. While unvisited states may potentially offer high intrinsic rewards, the agent cannot recognize this due to a lack of prior experiences. As a result, novelty-based exploration often leads the agent to repeatedly exploit states with relatively higher novelty among the visited states. Waiting for the natural shift in the balance to stochastic sampling to explore unvisited states introduces delays and inefficiencies, as the agent may collect redundant experiences in high intrinsic reward states. Additionally, this repetitive behavior increases the risk of premature convergence to suboptimal regions, further limiting effective exploration. Efficient exploration requires a dynamic and effective balance between these strategies to mitigate such risks. These two strategies can overlap or interfere, making it harder for the agent to make effective decisions and adapt its exploration strategy.

In this paper, we propose KEA (Keeping Exploration Alive) to address the inefficiencies arising from the complexities of balancing the exploration-exploitation trade-off when combining SAC with novelty-based methods. KEA proactively coordinates different exploration strategies, producing a consistent exploration behavior by reducing the complexity of interactions between novelty-based exploration and exploration via stochastic policy. Specifically, we introduce an additional co-behavior agent (denoted as $\mathcal{A}^B$) that works alongside SAC, which incorporates existing novelty-based methods for exploration (denoted as $\mathcal{A}^{SAC}$), facilitating smoother coordination between the strategies.

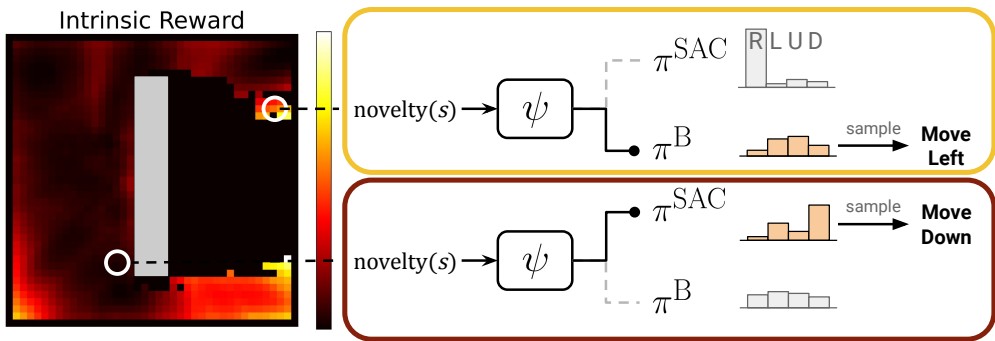

Figure 2: **Overview**. KEA introduces an additional co-behavior agent ($\mathcal{A}^{\text{B}}$) that works alongside and complements a novelty-based SAC agent ($\mathcal{A}^{\text{SAC}}$). A switching mechanism ($\psi$) proactively coordinates between $\mathcal{A}^{\text{SAC}}$ and $\mathcal{A}^{\text{B}}$ based on the current state novelty computed by the novelty-based model. The stochastic policies, $\pi^{\text{SAC}}$ and $\pi^{\text{B}}$, are derived from $\mathcal{A}^{\text{SAC}}$ and $\mathcal{A}^{\text{B}}$, respectively.

To implement proactive coordination between different exploration strategies, we introduce a switching mechanism based on state novelty, which dynamically shifts control between $\mathcal{A}^{\text{SAC}}$ and $\mathcal{A}^{\text{B}}$. This allows the agent to maintain high stochasticity in high novelty regions until extrinsic rewards are obtained. By proactively coordinating $\mathcal{A}^{\text{SAC}}$ and $\mathcal{A}^{\text{B}}$, KEA prevents the agent from prematurely converging and revisiting novel regions without purpose. This coordination ensures the agent can escape local optimal by maintaining diverse exploration behaviors and avoiding deterministic actions in areas with high novelty but low entropy. Additionally, KEA leverages off-policy RL, enabling data collection using multiple exploration and action policies. This allows us to use distinct exploration strategies (from $\mathcal{A}^{\text{SAC}}$ and $\mathcal{A}^{\text{B}}$) to gather diverse data from the environment.

We evaluate our method in two experimental settings (Section 3). First, we analyze a 2D navigation task with sparse rewards to study the underlying challenges of novelty-based exploration. Then, we test KEA on the DeepMind Control Suite (Tassa et al. (2018)) using sparse rewards in continuous control tasks. In the 2D navigation task, we demonstrate that KEA substantially improves learning efficiency by proactively coordinating exploration strategies. Under varying Update-to-Data (UTD) ratios, KEA consistently outperforms RND (Burda et al. (2018b)) and NovelD (Zhang et al. (2021)), highlighting its efficiency and robustness. Similarly, in the more challenging tasks from the DeepMind Control Suite, KEA improves performance over both RND and NovelD in three continuous control tasks.

Our main contributions are as follows: **(1)** We analyze a potential problem when combining SAC with novelty-based exploration, where the complexity of managing exploration-exploitation trade-off can lead to inefficient exploration. **(2)** We propose a method that proactively coordinates exploration strategies, significantly improving exploration efficiency and consistency. Our method is simple to integrate with existing novelty-based exploration methods, offering broad applicability.

## 2 METHOD

### 2.1 BACKGROUND

A Markov Decision Process (MDP) is represented by the state $s \in S$, action $a \in A$, transition function $\mathcal{T} : (s, a) \to s'$, reward function $r : S \times A \to \mathbb{R}$, and discount factory $\gamma$. The agent's goal is to find a policy $\pi \colon S \to A$ that maps the state $s_t$ to the action $a_t$ for maximizing the sum of expected rewards. In this paper, we consider a setup where the primary reward of interest (the "extrinsic" reward) is a sparse binary signal, supplemented by dense "intrinsic" rewards calculated by an intrinsic reward model. In KEA, we denote the overall reward for $\mathcal{A}^{\text{SAC}}$ at each time step $t$ as $r_t = \beta^{ext}\, r_t^{ext} + \beta^{int}\, r_t^{int}$, where $r_t^{ext}$ represents the extrinsic reward from the environment, $r_t^{int}$ is the intrinsic reward from novelty-based exploration model, and $\beta^{ext}$ and $\beta^{int}$ are scaling hyperparameters. The overall reward for $\mathcal{A}^{\text{B}}$ at each time step $t$ is $r_t = \beta^{ext}\, r_t^{ext}$, incorporating only the extrinsic reward.

## 2.2 Overview

As Fig. 2, we introduce a co-behavior agent ($\mathcal{A}^{\text{B}}$), which works alongside $\mathcal{A}^{\text{SAC}}$, providing a complementary exploration strategy to address inefficiencies caused by the complexity of exploration-exploitation trade-off. To coordinate $\mathcal{A}^{\text{SAC}}$ and $\mathcal{A}^{\text{B}}$, we devise a switching mechanism, denoted as $\psi$, which dynamically coordinates based on state novelty, measured by the novelty-based model.

In this paper, because Soft Actor-Critic (SAC) (Haarnoja et al. (2018); Christodoulou (2019)) has demonstrated significant success in continuous control tasks, we use it as the base RL agent and leverage Random Network Distillation (RND) (Burda et al. (2018b)) to compute intrinsic reward for exploration (denoted as $\mathcal{A}^{\text{SAC}}$). In an off-policy manner, we can collect transitions with multiple policies while training with another. This allows us to use distinct exploration strategies (e.g. $\mathcal{A}^{\text{SAC}}$ and $\mathcal{A}^{\text{B}}$) to gather diverse data from the environment.

## 2.3 Exploration Strategies

**Novelty-based Exploration.** Novelty-based exploration encourages the agent to focus on novel states within the explored region, increasing the chances of discovering previously unvisited areas. In this paper, we use SAC as the base RL agent and leverage RND to compute intrinsic rewards that guide this exploration (denoted as $\mathcal{A}^{\text{SAC}}$). Specifically, the SAC policy is updated to account for both extrinsic rewards (from the environment) and intrinsic rewards (based on novelty), we modify the Soft Bellman update target for the Q network in SAC (Haarnoja et al. (2018)) as shown below:

$$y_Q = (\boldsymbol{\beta^{ext}}\ \boldsymbol{r^{ext}} + \boldsymbol{\beta^{int}}\ \boldsymbol{r^{int}}) + \gamma\ (\min_{\theta'_{1,2}} Q_{\theta'_i}(s', a') - \alpha\ log\pi^{\text{SAC}}(\cdot|s')) \tag{1}$$

, where $\beta^{ext}$ and $\beta^{int}$ are scaling factors for extrinsic and intrinsic rewards. The $\alpha$ is the temperature parameter controlling the entropy regularization. The $r^{int}$ is an intrinsic reward computed based on the state novelty, which measures the prediction error of Random Network Distillation (RND), calculated as:

$$r_t^{int} = ||\hat{f}(s_t; \theta) - f(s_t)||^2 \tag{2}$$

, where $f : \mathcal{O} \to \mathbb{R}^K$ represents a randomly initialized target network that maps an observation $s_t$ to an embedding in $\mathbb{R}^K$, and $\hat{f} : \mathcal{O} \to \mathbb{R}^K$ is a predictor network trained via gradient descent to minimize the expected mean squared error (MSE) with the target network.

**Stochastic Policy via Co-behavior Agent.** We introduce an additional co-behavior agent (denoted as $\mathcal{A}^{\text{B}}$) that works alongside a SAC agent that incorporates existing novelty-based methods for exploration (denoted as $\mathcal{A}^{\text{SAC}}$), facilitating smoother coordination between the strategies. $\mathcal{A}^{\text{B}}$ includes a stochastic policy that maintains high variance by slowing down its gradient updates until extrinsic rewards are obtained, ensuring to have an exploration strategy that always has high stochasticity.

With $\mathcal{A}^{\text{B}}$ which maintains high variance in its actions, we can proactively coordinate $\mathcal{A}^{\text{SAC}}$ and $\mathcal{A}^{\text{B}}$ to prevent the agent from relying solely on natural shifts in exploration strategies caused by changes in entropy and intrinsic rewards. This coordination ensures a consistent escape from local minima by maintaining diverse exploration behaviors and mitigating deterministic actions in regions of high novelty but low entropy.

In this paper, we implement $\mathcal{A}^{\text{B}}$ using another SAC agent to enhance data efficiency by sharing experiences in a unified replay buffer with $\mathcal{A}^{\text{SAC}}$. During training, experiences are sampled from this shared buffer, and the policy and Q-networks of $A^B$ are updated concurrently with those of $A^{SAC}$. The notable difference is that the co-behavior agent is trained using a different reward signal, only taking into account the primary (sparse reward) task, which allows it to retain high entropy for the exploration of unvisited states.

## 2.4 Switching Mechanism

Since our method involves two exploration strategies, we require a mechanism to determine when to use each. The role of the switching mechanism is crucial for proactively coordinating $\mathcal{A}^{\text{SAC}}$ and $\mathcal{A}^{\text{B}}$. Simply averaging the action distributions from both agent policies would not be effective, as their objectives may differ significantly. Instead, we design a switching mechanism that adapts based on the novelty of the agent's current state. This mechanism ensures that $\mathcal{A}^{\text{B}}$ operates near the

boundary between explored and unexplored regions, while $\mathcal{A}^{\text{SAC}}$ frequently revisits relatively novel states within the explored regions.

We define switching criterion as follows:

$$\pi(s_t) = \psi(r_t^{int}, \pi^{\text{SAC}}(s_t), \pi^{\text{B}}(s_t)), \tag{3}$$

$$\psi = \begin{cases} \pi^{\text{B}}(s_t) & \text{, if } r_t^{int} > \sigma \\ \pi^{\text{SAC}}(s_t) & \text{, otherwise} \end{cases} \tag{4}$$

where $\pi^{\text{B}}$ and $\pi^{\text{SAC}}$ are stochastic policies from $\mathcal{A}^{\text{B}}$ and $\mathcal{A}^{\text{SAC}}$, respectively, and $\sigma$ is a threshold hyperparameter. When the received intrinsic reward falls below the predefined threshold, the agent switches to $\mathcal{A}^{\text{SAC}}$ for novelty-based exploration, which encourages the agent to visit relatively novel areas more often. Conversely, when the received intrinsic reward exceeds the threshold, the agent switches to $\mathcal{A}^{\text{B}}$, focusing on stochastic policy exploration to enter unexplored regions. This switching mechanism provides a proactive coordination of exploration strategies, further improving the exploration efficiency.

## 3 Experiments

In this section, we evaluate the performance of KEA in several RL tasks with sparse rewards to demonstrate its ability to effectively manage the complex interactions between different exploration strategies and improve overall exploration efficiency. We begin by testing our method on a 2D Navigation task, where the agent must navigate to a fixed goal position while avoiding obstacles. Given that the Update-to-Data (UTD) ratio can significantly impact the exploration-exploitation trade-off, we next analyze how KEA manages these potential challenge and ensures consistent exploration under varying UTD ratios. Finally, we evaluate KEA on more challenging environments from the DeepMind Control Suite (Tassa et al. (2018)) with sparse rewards, which present additional difficulties for exploration in continuous control tasks.

For all experiments, we use Soft Actor-Critic (SAC) as the base RL agent and demonstrate the flexibility of our method by integrating it with two different novelty-based exploration methods. Specifically, we combine SAC with **Random Network Distillation (RND)** (Burda et al. (2018b)), denoted as **KEA-RND**, and also with **NovelD** (Zhang et al. (2021)), denoted as **KEA-NovelD**. Our method adapts these novelty-based approaches by incorporating a co-behavior agent ($\mathcal{A}^{\text{B}}$) and a dynamic switching mechanism ($\psi$) to proactively coordinate exploration strategies, ensuring more efficient and effective exploration.

Each method is evaluated across five random seeds. We present results as the mean and standard deviation of episodic return. The primary evaluation metric is mean episodic return, which reflects both task performance and convergence speed. Our results demonstrate that KEA significantly improves exploration efficiency by proactively coordinating exploration strategies, reducing the negative effect from the complexity of exploration-exploitation trade-off, and enhancing overall learning performance.

### 3.1 2D Navigation Task

**Task Description.** As shown in Fig. 3, the 2D Navigation task involves navigating an agent to a fixed goal position on the right (blue point) while avoiding an obstacle placed in the middle of the environment. The agent's starting position (green point) is randomly initialized within the left half of the environment, at the beginning of each episode. The environment provides sparse extrinsic rewards, meaning the agent only receives extrinsic rewards when it successfully reaches the goal.

The observation space is discrete, consisting of the agent's current *(x, y)* position, while the action space includes four possible actions: *(move right, move left, move up, move down)*. Additionally, the transition function is unknown, and the agent must learn to navigate the environment through trial and error. We implement this environment by Gymnasium (Towers et al. (2024)).

**Experimental Setup.** In this experiment, we compare the performance of our method (KEA-RND and KEA-NovelD) against standard SAC, RND, and NovelD, measured by the mean episodic return during training. The training is halted after the agent collects 300,000 transitions from the environment. Our method variants — KEA-RND and KEA-NovelD — use RND and NovelD, respectively,

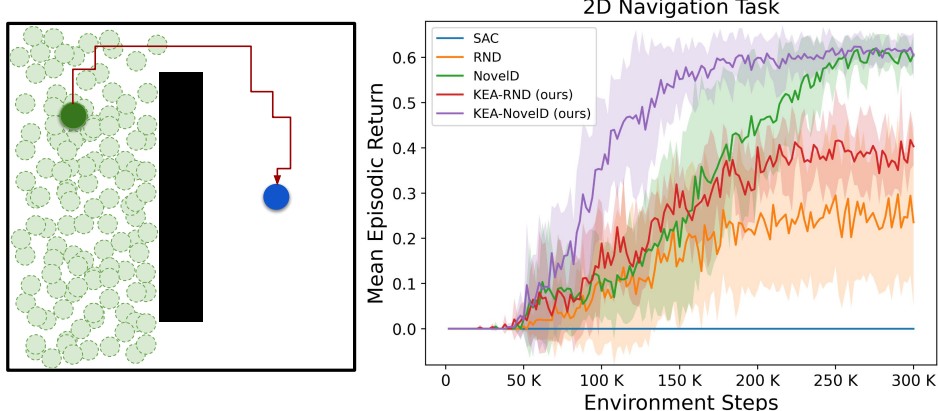

Figure 3: *Left*: 2D Navigation task involves navigating an agent from a randomly chosen start (light green circles) to a fixed goal position on the right (blue point) while avoiding an obstacle placed in the middle of the environment. *Right*: Mean episodic returns during training.

to compute the intrinsic rewards, combined with $\mathcal{A}^{\text{B}}$ and a dynamic switching mechanism to coordinate exploration strategies. Each method is tested across five random seeds, and we report both the mean and standard deviation of the performance to ensure statistical significance.

**Experimental Results.** As shown in Fig. 3, our method significantly outperforms the baselines. The final performances metrics are summarized in Table 1. KEA-RND achieves a mean episodic return of $0.403 \pm 0.042$ after 300,000 environment steps, compared to RND's $0.235 \pm 0.184$, representing a more than 70% improvement in performance. In the NovelD setup, NovelD reaches a mean episodic return of $0.607 \pm 0.042$, while KEA-NovelD achieves $0.604 \pm 0.051$ af-

| Method | Mean Return | STD |
|---|---|---|
| SAC | 0. | 0. |
| RND | 0.235 | 0.184 |
| KEA-RND (ours) | **0.403** | 0.042 |
| NovelD | **0.607** | 0.042 |
| KEA-NovelD (ours) | 0.604 | 0.051 |

Table 1: Mean Episodic Return in 2D Navigation task

ter 300,000 environment steps. Although the final performance between KEA-NovelD and NovelD is similar, KEA-NovelD converges significantly faster, reaching a return of 0.6 around 190,000 environment steps, whereas NovelD requires 250,000 steps to achieve a similar return. This demonstrates that our method not only maintains exploration efficiency but also improves convergence speed.

## 3.2 ANALYSIS OF EXPLORATION-EXPLOITATION TRADE-OFF

This trade-off is not only affected by the task and environment design but also influenced by how aggressively the SAC agent and intrinsic reward model are updated. Furthermore, the Update-to-Data (UTD) ratio affects the evolution of both entropy and intrinsic rewards, thereby impacting the shifting between these exploration strategies.

To evaluate KEA's ability to coordinate different exploration strategies and mitigate the inefficiency caused by the complexity of exploration-exploitation trade-off, we conducted an experiment with varying UTD ratios in the 2D Navigation task (shown in Fig. 3). We compare **KEA-RND** with **RND** to evaluate how different UTD ratios (for SAC and RND) affect the overall performances. This comparison highlights how KEA maintains efficient exploration and robustness across a range of UTD ratio settings. We further visualize the training process using a specific example to illustrate how the balance between exploration strategies shifts over time and how these shifts impact exploration performance. Our method demonstrates proactive coordination of exploration strategies, reducing inefficiencies from combining SAC with novelty-based methods, ensuring more consistent and efficient exploration.

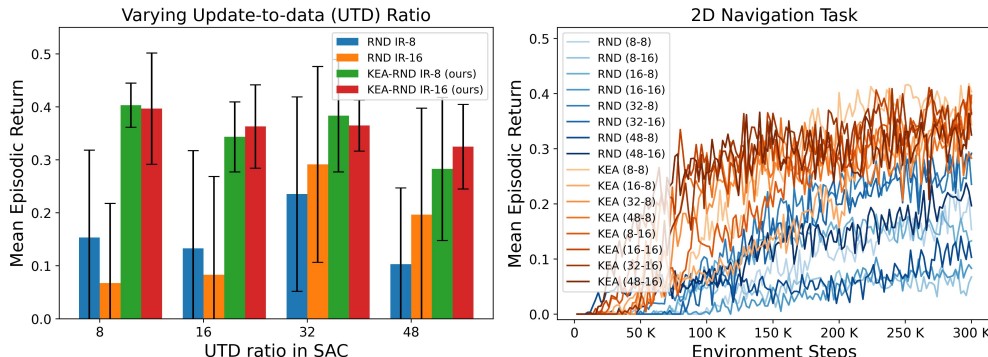

Figure 4: We test multiple UTD ratios for SAC at **8, 16, 32, and 48**, and for RND at **8 and 16**. The goal is to observe how the mean episodic return evolves during training. *Left*: The final results, where IR-8 and IR-16 represent UTD ratios in RND of 8 and 16, respectively. *Right*: The training curves for all UTD ratios, denoted as **SAC's-RND's** (e.g., 8-16 means the UTD ratio in SAC is 8 and in RND is 16). KEA consistently demonstrates better performance and a lower dependence on the chosen UTD ratio.

**Varying UTD Ratios.** In this experiment, we varied the UTD ratios by adjusting the number of SAC gradient updates to **8, 16, 32, and 48 times** per transition, while the number of RND updates was set to either **8 or 16 times**. The goal is to observe how the mean episodic return evolves during training, with a total of 300,000 samples collected from the environment.

As shown in the Fig. 4, KEA-RND consistently achieves higher mean episodic returns across all UTD ratios when compared to RND. Specifically, KEA-RND attains its best performance at $0.403 \pm 0.042$, whereas RND reaches a lower episodic return of $0.292 \pm 0.197$. However, at the highest UTD ratio (**48 updates**), both methods experience a performance decline. Despite this drop, KEA-RND maintains a better performance advantage over RND. Moreover, KEA-RND exhibits smaller standard deviations across all configurations, indicating that it is more robust and stable even as the update intensity increases.

**Visualization.** In Fig. 7, we visualize intrinsic rewards, entropy, and action probabilities throughout the training process to illustrate how exploration evolves over time. While RND successfully reaches the goal in its best cases for both 48 and 8 gradient updates ((a2) and (b2)), it becomes stuck in local minima in the worst cases ((a1) and (b1)), limiting further exploration. In contrast, KEA-RND consistently reaches the goal across all setups. Compared to RND, KEA-RND maintains higher entropy in regions with high intrinsic rewards, especially before reaching the goal. This demonstrates that our method proactively coordinates different exploration strategies (novelty-based exploration via $\mathcal{A}^{SAC}$ and stochastic policy via $\mathcal{A}^{B}$), thereby reducing negative effect from the complexity of exploration-exploitation trade-off. As a result, KEA-RND ensures more thorough exploration, decreasing the likelihood of getting stuck in local minima.

### 3.3 DEEPMIND CONTROL SUITE

**Task Description.** The DeepMind Control Suite (Tassa et al. (2018)) is a set of continuous control tasks to evaluate RL algorithms. These tasks simulate various physical environments and require agents to learn complex motor skills to achieve specified goals. Observation spaces are continuous, consisting of joint positions and velocities, while action spaces are represented as continuous values (e.g., joint torques or forces). The number of observation and action dimensions depends on the specific task.

**Experimental Setup.** In this experiment, we compare the performance of our method (KEA-RND and KEA-NovelD) against standard SAC, RND, and NovelD, measured by the mean episodic reward during training. The training is halted after the agent collects 500,000 transitions from the environment. As describe earlier in 3.1, KEA-RND and KEA-NovelD incorporate a co-behavior agent ($\mathcal{A}^{B}$) and a dynamic switching mechanism to proactively coordinate exploration strategies. They use RND and NovelD, respectively, to compute intrinsic rewards. Each method is tested across five

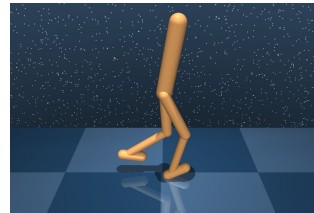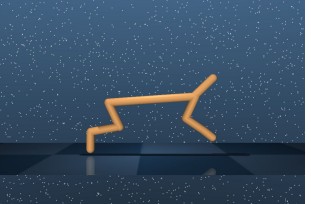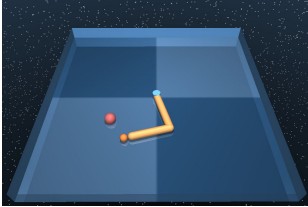

Figure 5: Three tasks from the DeepMind Control Suite (Tassa et al. (2018)) are used in this paper: **Walker Run**, **Cheetah Run**, and **Reacher Hard**. The objective in the first two tasks is to run as fast as possible, while in the third task, the agent must reach a specified goal position (represented by a red dot).

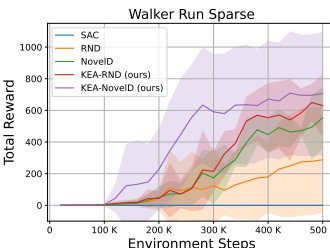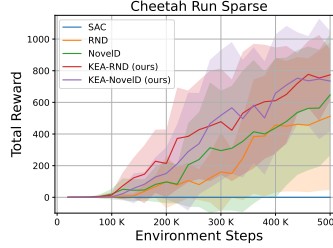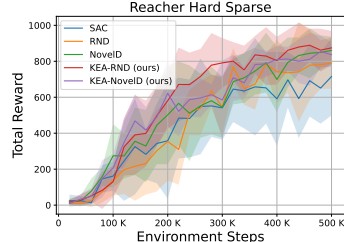

Figure 6: Performance on three continuous control tasks from the DeepMind Control Suite. Our method (KEA-RND and KEA-NovelD) performs notably better than baselines in more challenging exploration tasks. The shaded regions indicate the standard deviation across evaluation runs.

random seeds, and we report both the mean and standard deviation of the performance to ensure statistical significance.

We evaluate the methods on three tasks from the DeepMind Control Suite: **Walker Run Sparse**, **Cheetah Run Sparse**, and **Reacher Hard Sparse** (shown in Fig. 5). In Reacher Hard Sparse, the reward structure is originally sparse. For Walker Run Sparse and Cheetah Run Sparse, rewards are provided sparsely only when the original reward (from DeepMind Control Suite) exceeds a certain threshold. The threshold for Walker Run is set at 0.3, while for Cheetah Run, it is 0.35.

**Experimental Results.** As shown in Fig. 6, Walker Run Sparse and Cheetah Run Sparse present significant challenges for exploration. Without novelty-based exploration, SAC struggles to reach the goal of these tasks. In contrast, Reacher Hard Sparse is relatively easier, as SAC can reach the goal even without intrinsic rewards. Besides, the addition of novelty-based exploration improves performance across all three tasks, and our method further enhances this performance.

As shown in Table 2, after 500,000 environment steps, KEA-RND achieves significant improvements over RND, with increases of 119%, 51%, and 11% in mean episodic rewards across the three tasks. Similarly, KEA-NovelD demonstrates approximately a 10% improvement over NovelD. Although KEA-NovelD shows similar results to NovelD on the Reacher Hard Sparse task, it performs notably better performance in the more challenging exploration tasks, Walker Run Sparse and Cheetah Run Sparse.

| Method | Walker Run Sparse | | Cheetah Run Sparse | | Reacher Hard Sparse | |
|---|---|---|---|---|---|---|
| | Mean | STD | Mean | STD | Mean | STD |
| SAC | 0. | 0. | 0. | 0. | 715.17 | 216.57 |
| RND | 287.65 | 334.12 | 512.02 | 466.26 | 790.32 | 143.26 |
| KEA-RND (ours) | **629.74** | 196.75 | **773.76** | 162.74 | **874.61** | 94.58 |
| NovelD | 553.26 | 191.03 | 647.29 | 382.58 | **860.40** | 76.15 |
| KEA-NovelD (ours) | **706.47** | 389.23 | **734.67** | 316.95 | 837.12 | 68.95 |

Table 2: Mean Episodic Return on three tasks from the DeepMind Control Suite.

## 4 RELATED WORK

Computing novelty to improve exploration has emerged as a critical component for improving exploration efficiency in sparse reward environments, where extrinsic rewards are limited (Ladosz et al. (2022); Burda et al. (2018a); Kim et al. (2018)). These methods complement our work, as KEA can integrate with various curiosity- and novelty-based explorations.

**Prediction Error-based Novelty.** One popular approach is prediction error-based novelty, which measures state novelty by predicting the next state and calculating the error. Stadie et al.(Stadie et al. (2015)) compute the error between the predicted and the actual state in the latent space, while ICM(Pathak et al. (2017)) measures the prediction error of an agent's ability to anticipate action outcomes in a learned feature space using a self-supervised inverse dynamics model. RND (Burda et al. (2018b)) computes state novelty using prediction error of a randomly initialized network.

**Count-based Novelty.** Count-based novelty methods offer another effective strategy by measuring the novelty based on state visitation frequency. Early works (Bellemare et al. (2016); Ostrovski et al. (2017); Tang et al. (2017)) use pseudo-counts to estimate state visitation in high-dimensional environments. Machado et al. (Machado et al. (2020)) improve upon earlier methods by using the norm of the successor representation for implicit state counts without requiring domain-specific density models.

**Including Episodic Memory.** Some approaches combine episodic memory and life-long novelty. For example, NGU (Badia et al. (2020)) encourages exploration across episodes and over the agent's entire training process. RIDE (Raileanu & Rocktäschel (2020)) combines forward and inverse dynamics models with episodic count-based novelty to compute intrinsic rewards based on the distance between consecutive observations in the state embedding space. AGAC (Flet-Berliac et al. (2021)) combines episodic count-based novelty and the KL-divergence between the agent's policy and an adversarial policy to compute intrinsic rewards.

NovelD (Zhang et al. (2021)) integrates count-based novelty and novelty difference to encourage uniform and boundary exploration, showing strong results in sparse reward tasks. In this paper, we propose KEA and leverage NovelD to compute intrinsic rewards while introducing a co-behavior agent ($\mathcal{A}^{B}$) and a switching mechanism ($\psi$) to proactively coordinate exploration strategies and improve exploration efficiency.

**Other Exploration Methods.** Beyond prediction and count-based novelty approaches, other exploration methods include adding noise to parameters (Fortunato et al. (2017); Plappert et al. (2017)), computing intrinsic rewards via hierarchical reinforcement learning(Kulkarni et al. (2016)), using curriculum learning to guide exploration(Bengio et al. (2009); Portelas et al. (2020)), combining self-supervised reward-shaping methods and count-based intrinsic reward (Devidze et al. (2022)), using distance-based metrics for reward shaping (Trott et al. (2019)), diversifying policies by regularizing the loss function with distance metrics (Hong et al. (2018)), and combining a novelty-based exploration method with switching controls to determine which states to add shaping rewards in a multi-agent RL framework (Zheng et al. (2021)).

## 5 CONCLUSION

In this paper, we present **KEA**, a novel approach to addressing exploration challenges in sparse reward reinforcement learning. By introducing a co-behavior agent ($\mathcal{A}^{B}$) that works alongside SAC, which incorporates existing novelty-based methods, like RND and NovelD, for exploration ($\mathcal{A}^{SAC}$). KEA proactively coordinates exploration strategies through a dynamic switching mechanism. This ensures consistent discovery of new regions while maintaining a balance between exploration and exploitation. Compared to previous methods that rely solely on intrinsic rewards, KEA reduces the complexity of interactions between novelty-based exploration strategy and stochastic policy exploration strategy, leading to more stable training dynamics. Our experiments on sparse reward tasks from the DeepMind Control Suite demonstrate KEA's substantial improvement over RND and NovelD, underscoring its effectiveness in balancing different exploration strategies. While KEA offers several advantages, one limitation is that it is restricted to off-policy learning, as the co-behavior agent shares experiences with the target policy. Despite this, KEA provides a more principled approach to balancing exploration and exploitation, advancing exploration in complex environments.

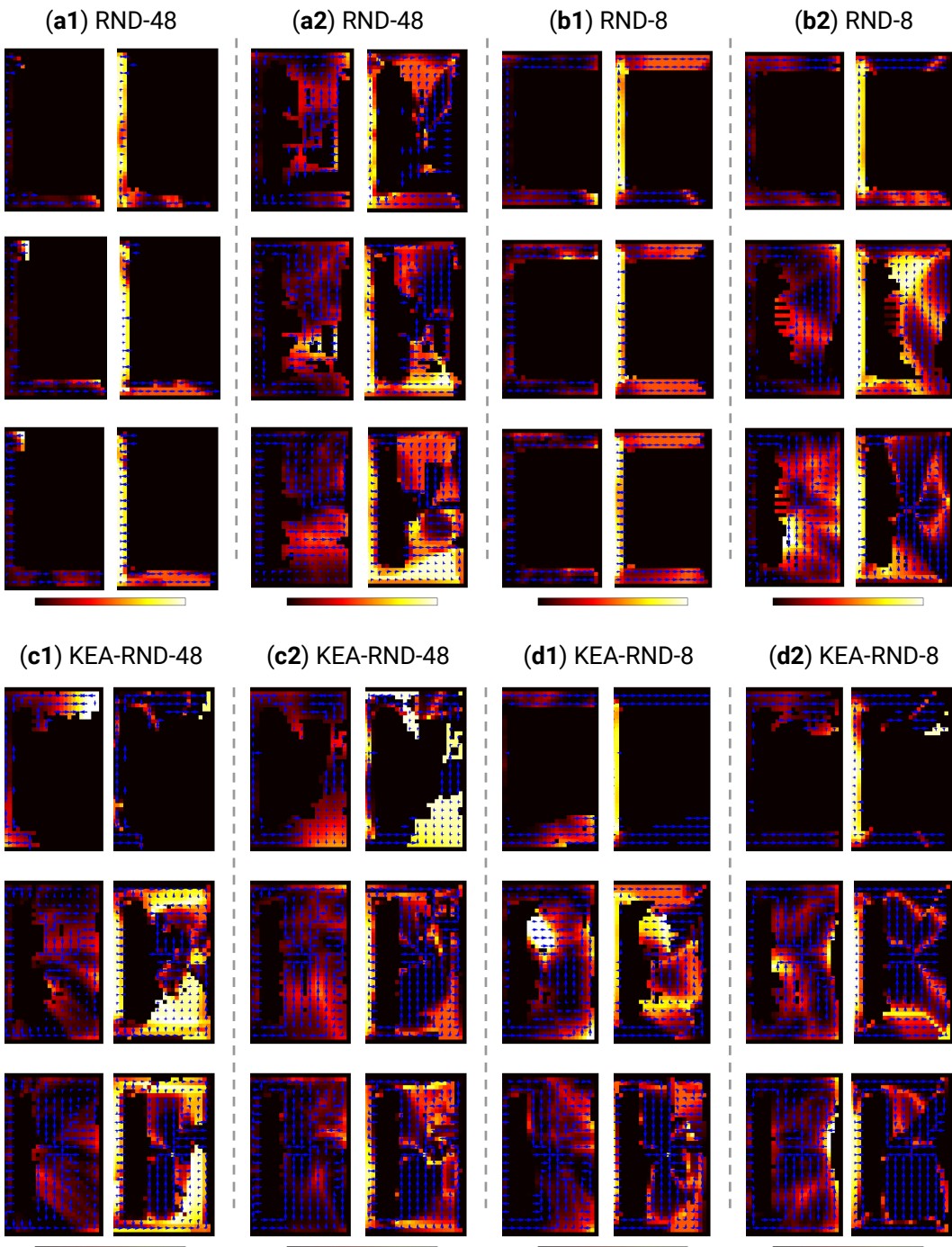

Figure 7: Panels (a) and (b) depict RND using 48 and 8 gradient updates, respectively, while panels (c) and (d) show KEA-RND under the same conditions. Additionally, (1) highlights the worst performance across five random seeds and (2) highlights the best. In each sub-figure (e.g., (a1)), intrinsic rewards (left) and entropy (right) are presented at three different stages of training: after collecting 20,000, 100,000, and 300,000 samples. Action probabilities are represented by arrows pointing in different directions. For clarity, we focus on the right part of the environment, which showcases the most interesting exploration behaviors, with unexplored states removed. The central obstacle in the environment is shown in Fig. 3.

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

# A    DIFFERENT SWITCHING THRESHOLDS

To analyze the sensitivity of KEA to different switching thresholds($\sigma$), we evaluate KEA-RND's mean episodic return on 2D Navigation task (3.1). The results, summarized in Table 3, show that while varying the switching threshold affects KEA's performance, all tested configurations consistently outperform RND ($0.292 \pm 0.197$). This demonstrates that KEA maintains robust performance across a reasonable range of threshold values.

To further investigate KEA's switching behavior between $\mathcal{A}^{\text{SAC}}$ and $\mathcal{A}^{\text{B}}$, we record their usage in Table 4. As the switching threshold ($\sigma$) increases, the usage of $\mathcal{A}^{\text{B}}$ decreases, as it is applied only in states with very high intrinsic rewards and tends to switch back to $\mathcal{A}^{\text{SAC}}$.

| Switching threshold | Mean Episodic Return | Standard |
|---|---|---|
| 0.50 | 0.358455 | 0.151244 |
| 0.75 | 0.348024 | **0.033442** |
| 1.00 | **0.407033** | 0.055562 |
| 1.25 | 0.348026 | 0.149555 |
| 1.50 | 0.333507 | 0.166823 |

Table 3: Evaluation of KEA-RND in different switching thresholds.

| Switching threshold | Ratio of using $\mathcal{A}^{\text{SAC}}$ | Ratio of using $\mathcal{A}^{\text{B}}$ |
|---|---|---|
| 0.50 | 0.7619 | 0.2381 |
| 0.75 | 0.8128 | 0.1872 |
| 1.00 | 0.8628 | 0.1372 |
| 1.25 | 0.8916 | 0.1084 |
| 1.50 | 0.9199 | 0.0800 |

Table 4: The ratio of using $\mathcal{A}^{\text{SAC}}$ and $\mathcal{A}^{\text{B}}$ in different switching threshold.

# B    A DIFFERENT SWITCHING MECHANISM

To evaluate the effectiveness of our proposed switching mechanism, we tested an alternative design in the 2D Navigation task, with RND as the novelty-based exploration strategy. This alternative mechanism, referred to as *KEA-RND-inv*, inverts KEA's original design: the agent switches to $\mathcal{A}^{\text{SAC}}$ in high intrinsic reward regions and to $\mathcal{A}^{\text{B}}$ in low intrinsic reward regions.

The results, summarized in Table 5, demonstrate that KEA's original switching mechanism achieves a higher mean episodic return and a lower standard deviation compared to the inverted design. These findings highlight the superior effectiveness of KEA's approach in coordinating exploration strategies.

| Method | Switching Mechanism | Mean Episodic Return | Standard |
|---|---|---|---|
| RND | – | 0.2354 | 0.1836 |
| KEA-RND | KEA's | **0.3835** | **0.1062** |
| KEA-RND-inv | Inverse KEA | 0.3186 | 0.1645 |

Table 5: Evaluation of a different switching mechanism.

