# OpenReview forum: "KEA: Keeping Exploration Alive by Proactively Coordinating Exploration Strategies in Curiosity-driven Exploration"
_ICLR.cc/2025/Conference — Submitted to ICLR 2025_

### Official Review · Reviewer_Ugcq · 2024-10-24

**Soundness:** 3
**Presentation:** 2
**Contribution:** 2
**Rating:** 5
**Confidence:** 4

**Summary:**

Reinforcement Learning tasks with sparse rewards are always hard to solve. Curiosity-based approaches are derived to facilitate the exploration. However, Soft Actor-Critic (SAC) does not perform well even with curiosity-based exploration. To address this issue, this paper introduces a new exploration strategy KEA (Keeping Exploration Alive), which combines entropy-based exploration and curiosity-based exploration strategies by introducing a simple switching mechanism between two policies with different variance. This mechanism decides which policy to deploy based on the intrinsic rewards.

For experiments, the authors concluded a simple 2D discrete navigation tasks and analysed the learned policy’s entropy and novelty. Results show that their method can improve performance or learning speed for NovelD and RND. They also tried the methods on a set of sparse continuous control tasks. KEA outperforms baseline algorithms. An ablation study of different UTD (Update-to-Data), KEA is more consistent than other baseline methods.

**Strengths:**

1. This paper addresses an important research questions. Applying Off-policy to tasks with sparse rewards is always tricky and existing curiosity-based methods are not working out of the box.
2. The paper is well motivated via a good visualization. The interplay between policy entropy and state novelty is interesting to look into.
3. The proposed algorithm is simple to implement with a minimal extra computing budget. The switching mechanism is clearly written.
4. The ablation study of various UTD ratios is also interesting.
5. Relevant literature is good summarised.

**Weaknesses:**

1. The scope of the paper is rather limited as the authors only tackle Soft Actor-Critic (SAC) in their paper. This paper can be extended to other off-policy algorithms as they all share the issues of premature convergence of the exploration and the influence from different UTD ratios.
2. The experiments are also not very convincing. First, the proposed 2D navigation task is too simple and the performance improvement is also not significant. Second, NovelD and RND both are used as baseline algorithms and NovelD has achieved better performance compared to RND. Yet, the latter experiments are all about RND. Thirdly, in the continuous tasks, the three tasks are not trained for enough time. The performance imporvement is also limited.
3. The presentation/clarity of the paper should also be improved. For example, I  didn't fully get Fig. 1's layout in the beginning. Some sentences are also written in a vague way without further explaination (See questions section). The training of the co-behavior policy is not mathematically defined.

**Questions:**

1. In Fig. 1, I didn't understand why "high intrinsic rewards can cause premature exploitation of novel regions". Could you further elaborate this sentence?
2. The statement of "Soft Actor-Critic (SAC) is known to be sensitive to the scale of rewards" should also be property justified either with a small experiment or some relevant literatures.
3. How is the co-behavior policy actually trained?
4. Does high stochasity always help with collecting diverse data from the environment? Methods such as novelD may encourage agent to try new actions to reach new states. This would consequently create more diverse data. But after introducing the high-variance co-behaviour policy, the agent may fall back to random policy and thus, collect repeated transitions.
5. Could you justify why RND is used for futher analysis not NovelD?

---

> ### Author Response · Authors · 2024-11-27
> **Response to Reviewer Ugcq (1/4)**
>
> Thank you for your engagement and constructive review of our manuscript! We would love to address each of your comments and questions.
>
> # Response to Weaknesses:
> > W1: The scope of the paper is rather limited as the authors only tackle Soft Actor-Critic (SAC) in their paper. This paper can be extended to other off-policy algorithms as they all share the issues of premature convergence of the exploration and the influence from different UTD ratios.
>
> **Response to W1**:
>
> Thank you for highlighting this point!
>
> KEA is indeed designed with SAC in mind, as SAC's ability to dynamically balance exploration and exploitation in off-policy settings provides a suitable foundation for our approach.
>
> For KEA to be applicable to other off-policy algorithms, there must be a similar interaction between the original exploration strategy and curiosity-based exploration. For example, in epsilon-greedy exploration, commonly used in Deep Q-Networks (DQN), actions are selected based on a fixed probability either through exploitation or random sampling. Since this exploration mechanism does not inherently interact with curiosity-based exploration, the balance-shifting behavior observed in SAC may not arise.
>
> Investigating KEA's applicability to other off-policy RL algorithms is an interesting and valuable direction. We have initiated preliminary experiments to explore this possibility, but additional time is required for setup and training. We will provide updated results as soon as they are available before the rebuttal ends.
>
>
> > W2: The experiments are also not very convincing. First, the proposed 2D navigation task is too simple and the performance improvement is also not significant. Second, NovelD and RND both are used as baseline algorithms and NovelD has achieved better performance compared to RND. Yet, the latter experiments are all about RND. Thirdly, in the continuous tasks, the three tasks are not trained for enough time. The performance imporvement is also limited.
>
> **Response to W2**:
>
> Thank you for the constructive feedback. We would love to address the concerns raised regarding our experiments and clarify the significance of our results.
>
> First, the results from the 2D navigation task are summarized in the table below. KEA-RND achieves more than a 70% improvement compared to RND. While the final performance of KEA-NovelD is comparable to that of NovelD, KEA-NovelD converges significantly faster, reaching a return of 0.6 around 190,000 environment steps, whereas NovelD requires approximately 250,000 steps to achieve a similar return.
>
> As we highlighted in **Response to Q1** in our response to the reviewer **tbHf**, the 2D navigation task is not as simple as it may seem. For instance, SAC with sparse rewards fails to converge even when the number of samples is doubled, highlighting the challenge of this environment when intrinsic reward signals are not effectively utilized.
>
> | Method     | Mean Episodic Return  | Standard  |
> | ---------- | --------------------- | --------- |
> | RND        | 0.235                 | 0.184	 |
> | NovelD     | 0.607                 | 0.042	 |
> | KEA-RND    | 0.403                 | 0.042	 |
> | KEA-NovelD | 0.604                 | 0.051	 |
>
> Second, regarding the choice of RND over NovelD for the analysis of varying UTD ratios, we selected RND due to its more straightforward exploration strategy, which enables clear and intuitive analyses and interpretations. Please refer to **Response to Q5** for further details.
>
> For the analysis of varying UTD ratios, we chose to use RND due to its more straightforward exploration strategy, which enables clear and intuitive analyses and interpretations. This simplicity helps isolate the specific impact of KEA without the added complexity introduced by NovelD. By using RND, we can more clearly interpret how KEA performs under different configurations, focusing on the core contributions of our method.
>
> Finally, we acknowledge the concern regarding the training time for the continuous control tasks. We have initiated experiments on the DeepMind Control Suite tasks with 1M environment steps to address this, but further time is required to complete them. We will provide updated results as soon as they are available before the rebuttal ends. Nonetheless, we emphasize that KEA demonstrates significantly faster learning compared to the baselines, particularly under limited demonstration steps.
>
> We hope this clarifies the contributions of our work and addresses your concerns.

---

> ### Author Response · Authors · 2024-11-27
> **Response to Reviewer Ugcq (2/4)**
>
> # Response to Weaknesses:
>
> > W3: The presentation/clarity of the paper should also be improved. For example, I didn't fully get Fig. 1's layout in the beginning. Some sentences are also written in a vague way without further explaination (See questions section). The training of the co-behavior policy is not mathematically defined.
>
> **Response to W3**:
>
> Thank you for your valuable suggestion regarding the clarity and presentation of our manuscript.
>
> First, the interpretation of Fig. 1 is explained in **Response to Q1**, where we clarify how curiosity-based exploration operates in SAC. Specifically, we discuss how it encourages the agent to explore states with high intrinsic rewards but may fail the agent to recognize truly unvisited states due to a lack of experience in the replay buffer. We will revise Fig. 1 to improve its clarity and alignment with the explanation provided.
>
> Second, as mentioned in **Response to Q2**, we will reference the original SAC paper [1] to support the statement about SAC’s sensitivity to reward scaling.
>
> Finally, as described in **Response to Q3**, we implement the co-behavior agent ($A^{B}$) using SAC, which shares a unified replay buffer with $A^{SAC}$ to enhance data efficiency. Both agents are trained concurrently by sampling experiences from the shared buffer.
>
> We will revise the manuscript with further explanation and ensure clarity.
>
> --------
> [1] Haarnoja, T., Zhou, A., Abbeel, P., & Levine, S. (2018, July). "Soft actor-critic: Off-policy maximum entropy deep reinforcement learning with a stochastic actor." In International conference on machine learning (pp. 1861-1870). PMLR.

---

> > ### Author Response · Authors · 2024-11-27
> > **Response to Reviewer Ugcq (3/4)**
> >
> > # Response to Questions:
> > > Q1: In Fig. 1, I didn't understand why "high intrinsic rewards can cause premature exploitation of novel regions". Could you further elaborate this sentence?
> >
> > **Response to Q1**:
> >
> > Thank you for the question.
> >
> > Curiosity-based exploration encourages the agent to explore states with high intrinsic rewards, which typically correspond to novel states. In SAC combined with curiosity-based exploration, these "novel" states refer to states that have already been visited and exhibit high novelty, rather than unvisited states. Although unvisited states may potentially offer high intrinsic rewards, this is not recognized by the agent because there are no experiences about them in the replay buffer. As a result, curiosity-based exploration guides the agent to repeatedly exploit those states with relatively higher novelty among the visited states.
> >
> > Exploration of truly unvisited states relies on the stochastic sampling provided by the policy in SAC. The agent dynamically shifts the balance between curiosity-based exploration and stochastic sampling based on intrinsic rewards and entropy. However, before this balance shifts, the agent may repeatedly collect similar experiences in the high intrinsic reward states, increasing the risk of premature convergence to those regions.
> >
> > We will attempt to clarify this in more detail in the revised manuscript.
> >
> > > Q2: The statement of "Soft Actor-Critic (SAC) is known to be sensitive to the scale of rewards" should also be property justified either with a small experiment or some relevant literatures.
> >
> > **Response to Q2**:
> >
> > Thank you for the suggestion. This statement is supported by Section 5.2 of the original SAC paper [1], which discusses the sensitivity of SAC to reward scaling. We will add this reference behind this statement and update our manuscript together with other suggestions.
> >
> >
> >
> > > Q3: How is the co-behavior policy actually trained?
> >
> > **Response to Q3**:
> >
> > As described in Section 2.3, we implement the co-behavior agent ($A^{B}$) using SAC. It shares a unified replay buffer with $A^{SAC}$ to enhance data efficiency. During training, experiences are sampled from this shared buffer, and the policy and Q-networks of $A^{B}$ are updated concurrently with those of $A^{SAC}$. The notable difference is that the co-behavior agent is trained using a different reward signal, only taking into account the primary (sparse) task, which allows it to retain high entropy for the exploration of unvisited states.
> >
> >
> > > Q4: Does high stochasity always help with collecting diverse data from the environment? Methods such as novelD may encourage agent to try new actions to reach new states. This would consequently create more diverse data. But after introducing the high-variance co-behaviour policy, the agent may fall back to random policy and thus, collect repeated transitions.
> >
> > **Response to Q4**:
> >
> > Thank you for the question.
> >
> > NovelD computes intrinsic rewards based on the novelty difference between two visited states, with episodic restrictions. However, as mentioned in **Response to Q1**, the novelty of unvisited states remains unrecognized by the agent because there are no experiences of these states in the replay buffer.
> >
> > In SAC combined with curiosity-based exploration, the exploration of truly unvisited states relies on stochastic sampling from the policy. Although the agent dynamically shifts the balance between curiosity-based exploration and stochastic sampling, this delay, along with the risk of repeatedly collecting similar experiences in states with high intrinsic rewards, can lead to inefficiencies.
> >
> > In KEA, we address this inefficiency by proactively switching to a high-stochasticity agent ($A^B$), without waiting for the balance to shift. While the high-variance co-behaviour policy may collect repeated transitions, its operation is limited to specific regions. The switching mechanism ensures a return to $A^{SAC}$ when entering lower intrinsic reward states. Our experiments demonstrate that KEA effectively improves performance when combining SAC with curiosity-based exploration.

---

> > > ### Author Response · Authors · 2024-11-27
> > > **Response to Reviewer Ugcq (4/4)**
> > >
> > > # Response to Questions:
> > >
> > >
> > > > Q5: Could you justify why RND is used for futher analysis not NovelD?
> > >
> > > **Response to Q5**:
> > >
> > > Thank you for the question.
> > >
> > > We have demonstrated KEA's performance with both RND and NovelD in our experiments. While NovelD achieves better performance than RND, its exploration strategy is significantly more complex, as it combines episodic counting-based bonuses with novelty differences to encourage exploration.
> > >
> > > For the analysis of varying UTD ratios, we chose to use RND due to its more straightforward exploration strategy, which enables clear and intuitive analyses and interpretations. This simplicity helps isolate the specific impact of KEA without the added complexity introduced by NovelD. By using RND, we can more clearly interpret how KEA performs under different configurations, focusing on the core contributions of our method.
> > >
> > > We acknowledge that additional experiments using NovelD in later analyses could provide a more comprehensive understanding of KEA’s impact in the context of state-of-the-art exploration strategies. We will prioritize this in future work to provide a more thorough evaluation and broader comparisons.

---

> > > > ### Comment · Reviewer_Ugcq · 2024-11-29
> > > >
> > > > First of all, I would thank the authors detailed response to my question which helps me understand the paper better. However I would like to maintain my score due to the limited impact of the paper and the significance of the performance improvement.
> > > >
> > > > I would kindly encourage the authors to expand their work in different RL algorithms, rather than SAC alone for a bigger impact. Algorithms such as PPO, which also involves a learnable std could already work as a good candidate and it's a on-policy algorithm. Secondly, the experiment setup should also improved. From my perspective, the proposed algorithm is a good candidate to even more challenging tasks, which involves also safety concerns and cannot be simply addressed by novelty-based methods. For example, in SafetyGymnasium [1], novelty-based method could easily counter problem of premature convergence as they keep violate the rules due to their exploration mechanism While KEA could work well as it could balance the "curiosity" and the actual goal.
> > > >
> > > > [1] Ji, J., Zhang, B., Zhou, J., Pan, X., Huang, W., Sun, R., Geng, Y., Zhong, Y., Dai, J., & Yang, Y. (2023). Safety-Gymnasium: A Unified Safe Reinforcement Learning Benchmark. ArXiv, abs/2310.12567.

---

> > > > > ### Author Response · Authors · 2024-12-03
> > > > > **Response to Reviewer Ugcq**
> > > > >
> > > > > Thank you for your feedback and for taking the time to understand our paper through the detailed responses.
> > > > >
> > > > > Our motivation for focusing KEA on off-policy RL methods stems from our exploration of combining curiosity- or novelty-based exploration with off-policy approaches for transfer learning across different continuous control tasks. During this process, we encountered challenges arising from the complex interactions between different exploration strategies.
> > > > >
> > > > > To address these limitations, we developed KEA to proactively coordinate exploration strategies, enabling more effective management of these interactions and improving overall performance in off-policy RL settings.

---

> ### Author Response · Authors · 2024-12-03
> **The Results in Response to W1 and W2**
>
> **The Results in Response to W1**:
>
> To explore KEA's generalization beyond SAC, we extended KEA to DQN and evaluated its performance on the 2D Navigation task.
>
> The preliminary results indicate that KEA does not improve the performance of DQN when combined with RND in this task. The results are summarized in the table below:
>
> | Method      | Mean    | STD     |
> | ----------- | ------- | ------- |
> | RND-DQN     | 0.6103  | 0.0240  |
> | KEA-RND-DQN | 0.5798  | 0.0164  |
>
> As discussed in **Response to W1**, KEA addresses potential problems arising from the interaction between different explorations. However, epsilon-greedy exploration, commonly used in DQN, does not interact significantly with novelty-based exploration. Instead, it either takes random action or selects the optimal action (considering the intrinsic reward).
>
> As part of future work, extending KEA to other off-policy methods, such as DDPG, TD3, or NAF, represents an interesting direction to further explore its generalization capabilities.
>
> ---
>
> **The Results in Response to W2**:
>
> The results of the extended training for the DeepMind Control Suite tasks to evaluate convergence at 1M steps are summarized in the table below.
>
>
> - Walker Run Sparse
>
> | Method      | Mean    | STD     |
> | ----------- | ------- | ------- |
> | NovelD      | 577.38  | 183.80  |
> | KEA-NovelD  | 846.66  |  98.86  |
>
> - Cheetah Run Sparse
>
> | Method      | Mean    | STD     |
> | ----------- | ------- | ------- |
> | NovelD      | 639.26  | 377.37  |
> | KEA-NovelD  | 870.20  | 162.94  |
>
> - Reacher Hard Sparse
>
> | Method      | Mean    | STD     |
> | ----------- | ------- | ------- |
> | NovelD      | 826.73  | 136.96  |
> | KEA-NovelD  | 872.36  |  55.87  |

---

### Official Review · Reviewer_vtHu · 2024-10-30

**Soundness:** 2
**Presentation:** 2
**Contribution:** 1
**Rating:** 5
**Confidence:** 4

**Summary:**

This paper proposed a switching mechanism to cooridnate exploration behaviors from two agents, an SAC agent with curiosity-based exploration strategy, and a co-behavior agent with only stochastic policy to draw actions. The proposed algorithm used the novelty intrinsic reward to choose which agent to output action by comaparing with a threshold. Therefore, the proposed algorithm can actively corredinate two different exploration strategies.

**Strengths:**

The approach presented in this paper is straightforward and easy to implement. It effectively combines two exploration strategies (curiosity-driven exploration and stochastic policy sampling). Experimental results demonstrate improved performance compared to approaches that rely on a single exploration strategy.

**Weaknesses:**

1. The underlying assumption for the switching criterion is not intuitive. In Equation (4) and the following explanation (from around Line 239 to Line 245), it is stated that for low intrinsic reward regions (i.e., low novelty), KEA switches to curiosity-based exploration to encourage the agent to explore novel areas. Conversely, for high novelty regions, KEA switches to a more stochastic sampling policy. However, it would make sense if it was the other way around, as high intrinsic rewards usually indicate that the area is novel and worth exploring further, making it more logical to apply a strong exploration strategy. Conversely, low intrinsic reward regions (low novelty) might be better suited for stochastic sampling. I recommend that this assumption be supported with more quantitative analysis or additional comparative experiments.

2. The paper conflates curiosity-based exploration and novelty-based intrinsic reward exploration. They are generally considered two distinct approaches for encouraging exploration. While this doesn't undermine the core idea of the paper, clearer distinctions between the two should be made, and the terminology used should be more precise. Including more related representative works would also help.

3. The experimental section can be improved. First, I suggest that the authors include additional baselines that focus specifically on exploration, beyond just the backbone algorithms. Since KEA’s core idea is to combine two exploration strategies, only comparing it against SAC, RND, and NovelD seems more like an ablation study (with only one agent in KEA involved). Representative SOTA works like [1][2][3] would offer better context for evaluating KEA’s performance. Additionally, for KEA itself, it would be valuable to explore other factors, such as how different values for the switching criterion (hyperparameter $\delta$) affect the model's switching behavior, how many times were $\pi^{SAC}$ and $\pi^B$ activated in training, etc.

4. The writing and expression can be improved, for example: (1) Lines 199-203 and Lines 128-130 are repeated verbatim, and (2) paragraphs like Lines 308-315 and Lines 404-411 could be better presented in table format for clearer understanding.

[1] Devidze, Rati, Parameswaran Kamalaruban, and Adish Singla. "Exploration-guided reward shaping for reinforcement learning under sparse rewards." Advances in Neural Information Processing Systems (2022).

[2] Trott, Alexander, et al. "Keeping your distance: Solving sparse reward tasks using self-balancing shaped rewards." Advances in Neural Information Processing Systems (2019).

[3] Hong, Zhang-Wei, et al. "Diversity-driven exploration strategy for deep reinforcement learning." Advances in neural information processing systems (2018).

**Questions:**

1. In Figure 2 (right-upper part), when the intrinsic reward is higher than a threshold, KEA switches to the co-behavior agent $\pi^B$. While this region is novel (i.e., high intrinsic reward), meaning the agent has not been well trained here. Why does the non-selected agent $\pi^{SAC}$ show an uneven distribution? Generally, for unfamiliar samples (those that have been rarely trained), the policy tends to output a relatively uniform distribution.

2. In Section 3.2, how is UTD set for both $\pi^{SAC}$ and $\pi^B$ respectively? In the UTD experiment, how is the batch size set? Since batch size has an impact on UTD performance, more details would be helpful.

3. The reference paper [4] presents a framework similar to KEA, where an additional agent is trained, and a switching control mechanism is designed. I encourage the authors to discuss this relevant work and make comparisons.

[4] Mguni, David, et al. "Learning to shape rewards using a game of two partners." Proceedings of the AAAI Conference on Artificial Intelligence. 2023.

I would like to increase the score if these concerns are addressed.

---

> ### Author Response · Authors · 2024-11-27
> **Response to Reviewer vtHu (1/4)**
>
> Thank you for your engagement and constructive review of our manuscript! We would love to address each of your comments and questions.
>
> # Response to Weaknesses:
> > W1: The underlying assumption for the switching criterion is not intuitive. In Equation (4) and the following explanation (from around Line 239 to Line 245), it is stated that for low intrinsic reward regions (i.e., low novelty), KEA switches to curiosity-based exploration to encourage the agent to explore novel areas. Conversely, for high novelty regions, KEA switches to a more stochastic sampling policy. However, it would make sense if it was the other way around, as high intrinsic rewards usually indicate that the area is novel and worth exploring further, making it more logical to apply a strong exploration strategy. Conversely, low intrinsic reward regions (low novelty) might be better suited for stochastic sampling. I recommend that this assumption be supported with more quantitative analysis or additional comparative experiments.
>
> **Response to W1**:
>
> Thank you for your insightful comment. We appreciate the opportunity to clarify the rationale behind our switching criterion and address your concerns.
>
> To evaluate the reviewer's suggestion, we implemented a new switching mechanism that inverses KEA's design. Specifically, in the reviewer's proposed mechanism, the agent switches to $A^{SAC}$ in high intrinsic reward regions and to $A^{B}$ in low intrinsic reward regions. The experimental results for this alternative design are presented in the table below.
>
> **Curiosity-Based Exploration**
> Curiosity-based exploration encourages the agent to explore states with high intrinsic rewards, which usually indicate novel states. However, in SAC combined with curiosity-based exploration, these "novel" states refer to states that have already been visited and exhibit high novelty, rather than unvisited states. Unvisited states, while potentially offering high intrinsic rewards, are not recognized by the agent because there are no experiences about them in the replay buffer. As a result, curiosity-based exploration often guides the agent to repeatedly exploit those states with relatively higher novelty among the visited states.
>
> **Rationale Behind KEA’s Switching Mechanism**
> Exploration of truly unvisited states relies on the stochastic sampling provided by the policy in SAC. Although the agent dynamically shifts the balance between curiosity-based exploration and stochastic sampling, this delay, along with the risk of repeatedly collecting similar experiences in states with high intrinsic rewards, can lead to inefficiencies.
> To address this inefficiency, KEA proactively switches to a high-stochasticity agent ($A^B$) in high intrinsic reward regions without waiting for the balance to naturally shift.
>
> **Challenges with the Reviewer’s Suggested Mechanism**
> When using the reviewer's proposed mechanism, the inefficiencies mentioned above still persist. Moreover, $A^{B}$ doesn't consider intrinsic reward. It will perform stochastic actions in the low intrinsic reward states before reaching the goal. These may decrease the learning efficiency.
>
> **Experimental Comparison**
> We tested the reviewer's suggested switching mechanism (denoted as *KEA-RND-inv*) in the 2D Navigation task, using RND as the curiosity-based exploration strategy for both mechanisms. The results, shown below, demonstrate that KEA's switching mechanism outperforms the reviewer's mechanism, achieving a higher mean episodic return and reduced standard deviation.
>
> | Method      | Switching Mechanism | Mean Episodic Return  | Standard  |
> | ----------- | ------------------- | --------------------- | --------- |
> | RND         | --                  | 0.235436              | 0.183551  |
> | KEA-RND     | KEA's               | **0.383489**              | **0.106154**  |
> | KEA-RND-inv | Reviewer's          | 0.318553              | 0.164527  |

---

> > ### Author Response · Authors · 2024-11-27
> > **Response to Reviewer vtHu (2/4)**
> >
> > # Response to Weaknesses:
> >
> >
> > > W2: The paper conflates curiosity-based exploration and novelty-based intrinsic reward exploration. They are generally considered two distinct approaches for encouraging exploration. While this doesn't undermine the core idea of the paper, clearer distinctions between the two should be made, and the terminology used should be more precise. Including more related representative works would also help.
> >
> > **Response to W2**:
> >
> > Thank you for the feedback. We appreciate the reviewer’s observation regarding the distinction between curiosity-based exploration and novelty-based intrinsic reward exploration.
> >
> > While some works (e.g. [4][5]) treat curiosity-based exploration as a broader concept that encompasses various exploration strategies, others (e.g. [2][6]) distinguish curiosity-based and novelty-based exploration by their underlying mechanisms. Specifically:
> > - Curiosity-based exploration rewards the agent for reducing uncertainty or prediction error about the environment.
> > - Novelty-based intrinsic reward exploration rewards the agent by computing a distance metric (e.g., in observation space or feature space) or state visitation counts to encourage exploration.
> >
> > We acknowledge that in some parts of the manuscript, we may not have clearly delineated these two approaches. We will revise the manuscript to clarify this distinction and ensure the terminology used is precise throughout.
> >
> >
> > > W4: The writing and expression can be improved, for example: (1) Lines 199-203 and Lines 128-130 are repeated verbatim, and (2) paragraphs like Lines 308-315 and Lines 404-411 could be better presented in table format for clearer understanding.
> >
> > **Response to W4**:
> >
> > Thank you for highlighting areas where the writing and presentation can be improved.
> >
> > 1. Repetition (Lines 199–203 and 128–130):
> > We appreciate your observation regarding the repeated text. We will revise the manuscript to remove redundancies while ensuring clarity and conciseness.
> >
> > 2. Table Presentation (Lines 308–315 and 404–411):
> > We agree that presenting the information in these sections as tables could improve clarity and readability. In the revised manuscript, we will reformat these paragraphs into table format to better organize and convey the data.
> >
> > We appreciate your detailed feedback and will implement these changes to improve the overall quality and presentation of the manuscript.

---

> > > ### Author Response · Authors · 2024-11-27
> > > **Response to Reviewer vtHu (3/4)**
> > >
> > > # Response to Weaknesses:
> > >
> > > > W3: The experimental section can be improved. First, I suggest that the authors include additional baselines that focus specifically on exploration, beyond just the backbone algorithms. Since KEA’s core idea is to combine two exploration strategies, only comparing it against SAC, RND, and NovelD seems more like an ablation study (with only one agent in KEA involved). Representative SOTA works like [1][2][3] would offer better context for evaluating KEA’s performance. Additionally, for KEA itself, it would be valuable to explore other factors, such as how different values for the switching criterion (hyperparameter
> > > ) affect the model's switching behavior, how many times were and activated in training, etc.
> > >
> > > **Response to W3**:
> > >
> > > **KEA’s Role in Advancing Novelty-based Intrinsic Reward Exploration**
> > > KEA’s core idea is to proactively coordinate two exploration strategies ---- novelty-based intrinsic reward exploration and stochastic sampling from SAC ---- rather than simply combining them. This proactive coordination addresses inefficiencies in exploration by dynamically switching these strategies based on intrinsic rewards and entropy signals.
> > >
> > > Our primary focus has been on analyzing KEA within the context of novelty-based intrinsic reward exploration. To provide a clearer and more comprehensive evaluation, we prioritized experiments with RND and NovelD.
> > > - RND provides a simple and intuitive baseline, allowing us to conduct clear analyses without additional complexity.
> > > - NovelD represents a state-of-the-art method in novelty-based intrinsic reward exploration, offering a fair comparison within this paradigm.
> > >
> > > While additional baselines such as [1][2][3] are indeed valuable for broader comparisons, this work specifically aims to advance novelty-based intrinsic reward methods for exploration. By evaluating KEA against other methods within this domain, we can isolate the contributions of our approach and ensure a focused and fair analysis. That said, we recognize the potential value of including baselines from other exploration paradigms. Such comparisons could provide additional insights into the relative strengths and weaknesses of different approaches. Incorporating these comparisons will be a priority in our future work, where we aim to explore the complementarity and trade-offs between these paradigms.
> > >
> > > We appreciate the suggestion and will include these papers in the related works to further complement the discussion of alternative approaches to addressing exploration challenges.
> > >
> > > **Sensitivity to Switching Thresholds**
> > > During the rebuttal, we conducted new experiments to evaluate KEA-RND under different switching thresholds ($\sigma$) using the 2D Navigation task. The results are summarized in the table below.
> > >
> > > 1. Performance Impact:
> > > While varying the switching threshold does affect KEA's performance, all tested configurations consistently outperform RND (0.292 \pm 0.197). This indicates that KEA maintains robust performance across a reasonable range of threshold values.
> > >
> > > | Switching threshold ($\sigma$) | Mean Episodic Return  | Standard  |
> > > | ------------------------------ | --------------------- | --------- |
> > > | 0.50                           | 0.358455              | 0.151244	 |
> > > | 0.75                           | 0.348024              | 0.033442	 |
> > > | 1.00                           | 0.407033              | 0.055562	 |
> > > | 1.25                           | 0.348026              | 0.149555	 |
> > > | 1.50                           | 0.333507              | 0.166823	 |
> > >
> > > 2. Usage of $A^{SAC}$ and $A^{B}$:
> > > The table below provides further details on the ratio of $A^{SAC}$ and $A^{B}$ usage. As the switching threshold ($\sigma$) increases, the usage of $A^{B}$ decreases because it is applied only in states with very high intrinsic rewards and tends to switch back to $A^{SAC}$ quickly.
> > >
> > > | Switching threshold ($\sigma$) | Ratio of using $A^{SAC}$ | Ratio of using $A^{B}$   |
> > > | ------------------------------ | ------------------------ | ------------------------ |
> > > | 0.50                           | 0.7619                   | 0.2381                   |
> > > | 0.75                           | 0.8128                   | 0.1872                   |
> > > | 1.00                           | 0.8628                   | 0.1372                   |
> > > | 1.25                           | 0.8916                   | 0.1084                   |
> > > | 1.50                           | 0.9199                   | 0.0800                   |

---

> > > > ### Author Response · Authors · 2024-11-27
> > > > **Response to Reviewer vtHu (4/4)**
> > > >
> > > > # Response to Questions:
> > > > > Q1: In Figure 2 (right-upper part), when the intrinsic reward is higher than a threshold, KEA switches to the co-behavior agent. While this region is novel (i.e., high intrinsic reward), meaning the agent has not been well trained here. Why does the non-selected agent show an uneven distribution? Generally, for unfamiliar samples (those that have been rarely trained), the policy tends to output a relatively uniform distribution.
> > > >
> > > > **Response to Q1**:
> > > >
> > > > Thank you for raising this critical point.
> > > >
> > > > Below, we explain why the non-selected agent ($A^{SAC}$) exhibits an uneven distribution in this scenario.
> > > >
> > > > **Influence of Intrinsic Rewards on Action Bias:**
> > > > Intrinsic rewards play a significant role in guiding the agent's exploration, especially in the absence of extrinsic rewards. When the agent encounters a novel region (indicated by high intrinsic reward), the intrinsic reward biases the policy toward specific actions that lead back to these high intrinsic reward states. This preference introduces unevenness in the action distribution, as actions that maximize the intrinsic reward are prioritized.
> > > >
> > > > **High Priorities in Replay Buffer:**
> > > > Experiences involving high intrinsic rewards usually have high priorities in the prioritized replay buffer. While these states may have been visited only a few times, their high priority ensures that they are sampled more frequently during training updates for SAC. This disproportionate sampling amplifies the influence of these experiences on the policy, further skewing the action distribution toward a non-uniform pattern.
> > > >
> > > >
> > > > > Q2: In Section 3.2, how is UTD set for both and respectively? In the UTD experiment, how is the batch size set? Since batch size has an impact on UTD performance, more details would be helpful.
> > > >
> > > > **Response to Q2**:
> > > >
> > > > Thank you for the question!
> > > >
> > > > In Section 3.2, the Update-to-Data (UTD) ratios for both $A^{SAC}$ and $A^{B}$ are set to the same values across experiments, specifically 8, 16, 32, and 48. Updates to $A^{B}$ are performed simultaneously with updates to $A^{SAC}$.
> > > >
> > > > Regarding the batch size, we use a fixed batch size of 512 for all tasks, experiments, and methods presented in the paper.
> > > >
> > > >
> > > > > Q3: The reference paper [4] presents a framework similar to KEA, where an additional agent is trained, and a switching control mechanism is designed. I encourage the authors to discuss this relevant work and make comparisons.
> > > > [4] Mguni, David, et al. "Learning to shape rewards using a game of two partners." Proceedings of the AAAI Conference on Artificial Intelligence. 2023.
> > > >
> > > > **Response to Q3**:
> > > >
> > > > Thank you for the feedback.
> > > >
> > > > ROSA[4] is a multi-agent RL framework that formulates the problem as a two-player zero-sum Markov Game, which differs fundamentally from KEA’s formulation. KEA is a single-agent RL framework that formulates the problem as a Markov Decision Process (MDP). While KEA employs two agents, $A^{SAC}$ and $A^{B}$, they operate within a single-agent setup, sharing experiences and dynamically switching based on the intrinsic reward threshold.
> > > >
> > > > ROSA presents a framework that includes two agents: a *shaper* and a *controller*. The *shaper* uses switching controls to determine which states to add shaping rewards, thereby enabling the *controller* to learn more efficiently by optimizing its policy with these additional rewards. In contrast, KEA’s additional agent ($A^{B}$) provides an alternative exploration strategy to complement $A^{SAC}$. KEA's switching mechanism switches between the two agents, rather than deciding whether to apply shaping rewards to specific states, as in ROSA.
> > > >
> > > > Although both frameworks employ two agents and a switching mechanism, their objectives and designs are distinct. KEA focuses on dynamically coordinating exploration strategies to address inefficiencies in combining SAC with curiosity-based exploration, while ROSA emphasizes selectively adding shaping rewards to guide policy learning.
> > > >
> > > > We appreciate the suggestion and will include a discussion of ROSA in the related works to further clarify KEA’s novelty and distinctions.
> > > >
> > > > -----
> > > > [1] Devidze, Rati, et al. "Exploration-guided reward shaping for reinforcement learning under sparse rewards." Advances in Neural Information Processing Systems (2022).
> > > >
> > > > [2] Trott, Alexander, et al. "Keeping your distance: Solving sparse reward tasks using self-balancing shaped rewards." Advances in Neural Information Processing Systems (2019).
> > > >
> > > > [3] Hong, Zhang-Wei, et al. "Diversity-driven exploration strategy for deep reinforcement learning." Advances in neural information processing systems (2018).
> > > >
> > > > [5] Zheng, Lulu, et al. "Episodic multi-agent reinforcement learning with curiosity-driven exploration." Advances in Neural Information Processing Systems 34 (2021): 3757-3769.
> > > >
> > > > [6] Zhang, T., et al. (2021). "Noveld: A simple yet effective exploration criterion." Advances in Neural Information Processing Systems, 34, 25217-25230.

---

> ### Comment · Reviewer_vtHu · 2024-12-02
>
> Thanks to the authors' response, as many of my questions have been answered, I’d like to increase the score to 5. However, I still believe there is significant room for improvement for the paper to reach the level of a well-rounded contribution. Therefore, I still consider this paper to be marginally below the acceptance threshold.

---

### Official Review · Reviewer_kGRm · 2024-10-31

**Soundness:** 2
**Presentation:** 3
**Contribution:** 2
**Rating:** 5
**Confidence:** 4

**Summary:**

The paper introduces KEA (Keeping Exploration Alive), a framework designed to enhance exploration efficiency in reinforcement learning, specifically for sparse reward environments. The core contribution of the paper is a novel method that combines the Soft Actor-Critic (SAC) approach with a co-behavior agent, designed to improve exploration by proactively coordinating multiple strategies. This coordination is facilitated by a dynamic switching mechanism, which alternates control between the SAC agent, which includes a curiosity-driven exploration method, and the co-behavior agent based on the novelty of the state being explored. This mechanism enables KEA to maintain high stochasticity in high-novelty regions and avoid premature convergence, thereby improving both learning efficiency and robustness.

**Strengths:**

- The method is easy to understand and contributes to alleviating an important issue in the field of reinforcement learning.

- Proposes a unique combination of SAC and a co-behavior agent with a dynamic switching mechanism.

- Shows substantial performance gains in benchmarks like 2D navigation tasks and robot control.

- Paper well written and easy to follow

**Weaknesses:**

- The paper should discuss how the switching mechanism threshold is tuned and its effect on the performance. This appears to be a key mechanism but somehow ablation studies on this component are limited.

- The evaluation lacks comparison with papers that address a similar issue, such as [Never give up: Learning directed exploration strategies, Badia et al., 2020]. I would suggest adding stronger baselines to validate the proposed method.

- The paper’s proposed method, KEA, is designed specifically with SAC in mind, leveraging SAC’s ability to handle exploration-exploitation trade-offs in off-policy settings. However, the authors do not discuss how KEA might generalize to other off-policy algorithms.

- Finally, one of my main concerns comes from the switching mechanism, which utilizes a fixed threshold to decide when to switch between the SAC agent with curiosity-driven exploration and the co-behavior agent. While this approach appears effective in simulations, using a fixed threshold might lack the flexibility needed for different environments or tasks, especially those with varying reward sparsity or novelty patterns. This inflexibility could limit KEA’s adaptability, potentially leading to less effective exploration in more complex or dynamic scenarios

**Questions:**

-How sensitive is the model to the switching threshold parameter?

- Was this threshold hand-tuned or optimized in some manner?

- What is the computational impact of using two agents in terms of memory and processing requirements?

- Does the model perform well in highly dynamic environments with frequently changing goals?

- Can the method applied to other RL algorithms, beyond SAC?

---

> ### Author Response · Authors · 2024-11-23
> **Response to Reviewer kGRm (1/2)**
>
> Thank you for your engagement and constructive review of our manuscript! We would love to address each of your comments and questions.
>
>
> # Response to Weaknesses:
> > W1: The paper should discuss how the switching mechanism threshold is tuned and its effect on the performance. This appears to be a key mechanism but somehow ablation studies on this component are limited.
>
> **Response to W1**:
>
> Thank you for the constructive feedback.
>
> We have conducted experiments to analyze the sensitivity of KEA to different switching thresholds, as detailed in **Response to Q1**. These experiments demonstrate how the threshold selection impacts performance, providing insights into its role in KEA's effectiveness.
>
> Additionally, in this paper, we hand-tuned the switching threshold based on preliminary experiments. Please refer to **Response to Q2** for further details.
>
>
> > W2: The evaluation lacks comparison with papers that address a similar issue, such as [Never give up: Learning directed exploration strategies, Badia et al., 2020]. I would suggest adding stronger baselines to validate the proposed method.
>
> **Response to W2**:
>
> Thank you for the valuable suggestion.
>
> NovelD [1] is often regarded as performing better than Never Give Up (NGU) for addressing hard exploration tasks [2]. Comparing with NovelD should be meaningful to demonstrate KEA's ability.
>
> On the other hand, NGU uses Recurrent Replay Distributed DQN (R2D2) [3], an RNN-based Deep Q-Network (DQN) algorithm. However, in this paper, we focus on combining KEA with Soft Actor-Critic (SAC).
> Exploring whether KEA can improve the performance of DQN combined with curiosity-based exploration is an interesting question. Please refer to **Response to Q5** for more details. Furthermore, understanding how RNN-based methods impact the interaction between different exploration strategies would be an intriguing line of investigation.
>
> That said, directly applying KEA to NGU is not straightforward, due to the differences in algorithmic structure and exploration strategies. This remains an open question and an exciting avenue for future work to investigate the interplay between RNN-based models and dual exploration strategies.
>
> > W3: The paper’s proposed method, KEA, is designed specifically with SAC in mind, leveraging SAC’s ability to handle exploration-exploitation trade-offs in off-policy settings. However, the authors do not discuss how KEA might generalize to other off-policy algorithms.
>
> **Response to W3**:
>
> Thank you for highlighting this point. KEA is indeed designed with SAC in mind, as SAC's ability to balance exploration and exploitation in off-policy settings provides a suitable foundation for our approach. Investigating KEA's applicability to other off-policy RL algorithms is an interesting and valuable direction.
>
> We have initiated preliminary experiments to explore this possibility, but additional time is required for setup and training. For further details on our thoughts regarding generalization to other RL algorithms, please refer to **Response to Q5**.
>
> > W4: Finally, one of my main concerns comes from the switching mechanism, which utilizes a fixed threshold to decide when to switch between the SAC agent with curiosity-driven exploration and the co-behavior agent. While this approach appears effective in simulations, using a fixed threshold might lack the flexibility needed for different environments or tasks, especially those with varying reward sparsity or novelty patterns. This inflexibility could limit KEA’s adaptability, potentially leading to less effective exploration in more complex or dynamic scenarios
>
> **Response to W4**:
>
> Thank you for the insightful feedback.
>
> The intrinsic rewards in KEA are normalized using a running mean and standard deviation, which ensures that their range is consistent across different environments and tasks. For instance, in the 2D Navigation task, we set the switching threshold to 1, while in the DeepMind Control Suite tasks, we consistently use a threshold of 0.75 across all tasks. This normalization helps mitigate the impact of varying reward scales and sparsity patterns.
>
> We agree that a dynamic threshold could potentially improve KEA's adaptability to environments with highly variable reward sparsity or novelty patterns. However, defining an effective dynamic switching mechanism, including identifying the appropriate signals to adjust the threshold, is a non-trivial challenge and an interesting direction for future research.

---

> ### Author Response · Authors · 2024-11-23
> **Response to Reviewer kGRm (2/2)**
>
> # Response to Questions:
> > Q1: How sensitive is the model to the switching threshold parameter?
>
> **Response to Q1**:
>
> We conducted experiments to analyze the sensitivity of KEA-RND to different switching thresholds, using the 2D Navigation task as the evaluation setting. The results are summarized in the table below.
>
> Our findings indicate that while varying the switching threshold does affect KEA's performance, all tested configurations consistently outperform RND (0.292 \pm 0.197). This suggests that KEA is robust to threshold variations within a reasonable range.
>
> (For reference, the intrinsic rewards are normalized using a running mean and standard deviation, resulting in a standard value of approximately 1.)
>
>
> | Switching threshold ($\sigma$) | Mean Episodic Return  | Standard  |
> | ------------------------------ | --------------------- | --------- |
> | 0.50                           | 0.358455              | 0.151244	 |
> | 0.75                           | 0.348024              | 0.033442	 |
> | 1.00                           | 0.407033              | 0.055562	 |
> | 1.25                           | 0.348026              | 0.149555	 |
> | 1.50                           | 0.333507              | 0.166823	 |
>
> > Q2: Was this threshold hand-tuned or optimized in some manner?
>
> **Response to Q2**:
>
> In this paper, the switching threshold was hand-tuned based on preliminary experiments. We hand-tuned the threshold by checking the usage of $A^{SAC}$ is around 85% to 90%.
>
> However, we acknowledge that automated hyperparameter tuning methods, such as grid search or Bayesian optimization, could be employed to systematically optimize the threshold.
>
> > Q3: What is the computational impact of using two agents in terms of memory and processing requirements?
>
> **Response to Q3**:
>
> As outlined in Section 2.3, the two agents utilize a unified replay buffer to share experiences, enhancing data efficiency and minimizing additional memory overhead. During the forward pass, the current intrinsic reward is computed to guide the switching mechanism in selecting which agent will estimate the action. Crucially, the action estimation process itself is equivalent to that of a single SAC agent, ensuring that the computational impact in terms of memory and processing requirements remains minimal.
>
>
> > Q4: Does the model perform well in highly dynamic environments with frequently changing goals?
>
> **Response to Q4**:
>
> Dynamic environments with frequently changing goals indeed pose additional challenges for exploration. However, the complex interaction between two exploration strategies ---- stochastic sampling and curiosity-based exploration ---- still exists when combining SAC with curiosity-based exploration.
>
> KEA improves performance primarily by addressing this challenge, enabling the agent to better explore unvisited states. In this paper, we focus on fixed-goal environments to provide clear and intuitive analyses and facilitate direct comparisons with baseline methods.
>
>
>
> > Q5: Can the method applied to other RL algorithms, beyond SAC?
>
> **Response to Q5**:
>
> Thank you for the interesting question.
>
> In this paper, we focus on the potential problems arising from the interaction between SAC's stochastic exploration and curiosity-based exploration. For KEA to be applicable to other RL algorithms, there must be a similar interaction between the original exploration strategy and curiosity-based exploration.
>
> For example, in epsilon-greedy exploration, commonly used in Deep Q-Networks (DQN), based on a fixed probability, actions are chosen either through exploitation or random sampling. Since this exploration mechanism does not interact with curiosity-based exploration, the balance-shifting behavior observed in SAC may not occur.
>
> Investigating KEA's applicability to other RL algorithms is an interesting direction. We have begun preliminary experiments in this area, but further time is required for setup and model training. We will provide updated results as soon as they are available before the rebuttal ends.
>
>
> ----
> [1] Zhang, T., Xu, H., Wang, X., Wu, Y., Keutzer, K., Gonzalez, J. E., & Tian, Y. (2021). "Noveld: A simple yet effective exploration criterion." Advances in Neural Information Processing Systems, 34, 25217-25230.
>
> [2] Wan, S., Tang, Y., Tian, Y., & Kaneko, T. (2023). "DEIR: efficient and robust exploration through discriminative-model-based episodic intrinsic rewards." arXiv preprint arXiv:2304.10770.
>
> [3] Kapturowski, S., Ostrovski, G., Quan, J., Munos, R., & Dabney, W. (2018, September). "Recurrent experience replay in distributed reinforcement learning." In International conference on learning representations.

---

> ### Author Response · Authors · 2024-12-03
> **The Results in Response to Q5**
>
> To explore KEA's generalization beyond SAC, we extended KEA to DQN and evaluated its performance on the 2D Navigation task.
>
> The preliminary results indicate that KEA does not improve the performance of DQN when combined with RND in this task. The results are summarized in the table below:
>
> | Method      | Mean    | STD     |
> | ----------- | ------- | ------- |
> | RND-DQN     | 0.6103  | 0.0240  |
> | KEA-RND-DQN | 0.5798  | 0.0164  |
>
> As discussed in **Response to Q5**, KEA addresses potential problems arising from the interaction between different explorations. However, epsilon-greedy exploration, commonly used in DQN, does not interact significantly with novelty-based exploration. Instead, it either takes random action or selects the optimal action (considering the intrinsic reward).
>
> As part of future work, extending KEA to other off-policy methods, such as DDPG, TD3, or NAF, represents an interesting direction to further explore its generalization capabilities.

---

### Official Review · Reviewer_tbHf · 2024-11-04

**Soundness:** 2
**Presentation:** 3
**Contribution:** 2
**Rating:** 5
**Confidence:** 4

**Summary:**

This paper proposes Keeping Exploration Alive (KEA), an algorithm that switches between exploration policies depending on the value of the intrinsic reward in order to reduce sub-optimal performance due to explore-exploit tradeoffs.

**Strengths:**

This paper is generally well written, and tackles a valid problem in RL, how to gracefully handle the explore-exploit tradeoff. The core results are clear, and the resulting algorithm seems to outperform the baseline by a significant amount on the evaluation task suite. The approach is, to my knowledge, original, though other methods have tackled this problem before in various ways.

**Weaknesses:**

The core issue I have with this paper is that there's very little exploration of the proposed algorithm beyond claims of superior performance. The core concept is simple (not a bad thing), but there's no real consideration given to what makes it good or useful. Seeing as the core contribution of this work is very simple, it should be straightforward to do deeper analysis and benchmarking, but this paper doesn't do much of that, and largely just concludes "it works."

As a result, I can't help but feel this paper is just sort of insubstantial and doesn't contribute that much to the field. It proposes an idea that is novel, but straightforward as an extension of past work, and doesn't do anything with the idea beyond show that it outperforms the baseline. If accepted as is, I'm not sure how much this will actually move the topic of exploration forward.

As such, I'm inclined to recommend rejection, but would like to see an expanded version of this work submitted to a future conference, because I do like the core idea of dynamically switching between explore and exploit policies for off-policy RL.

**Questions:**

Some additional questions and suggestions:

-In figure 3, the SAC line seems to be flat? I am surprised this task cannot be learned at all by SAC, and that the best method only reaches a ~60% success rate. Perhaps the degree of sparsity is not clear to me from the task description. Perhaps that sparsity could be elaborated on somehow?

-I suggest adjusting page spacing so that the critical algorithms 3 and 4 are on the same page, since they are connected and key to this paper's contribution

-Figure 4 feels like supplemental material to me, since it's basically just validating hyperparameter choices.

-The caption on figure 6 should say what the shaded regions represent.

-Also in figure 6, it looks like the algorithms haven't clearly converged on most of the three tasks. 500k environment steps is not a very large number, perhaps it would be worthwhile to run these experiments for the commonly-used 1 million steps to make sure there's separation at convergence?

-Why is it valuable to vary update to data ratios so much? Is there a reason this is a key hyperparameter for this algorithm? If so that should be explained in the paper. It's not clear how this relates to explore-exploit tradeoffs

-For that matter, why is sigma (which appears to be a key hyperparameter) not varied? It seems like performance should be sensitive to that parameter since it controls algorithm switching (the core contribution)

-Further, when and how much does KEA actually switch between the policies? This is the core phenomenon of this paper, so surely there should be some confirmation it's actually switching regularly and in a structured way, no?

---

> ### Author Response · Authors · 2024-11-26
> **Response to Reviewer tbHf (1/3)**
>
> Thank you for your engagement and constructive review of our manuscript! We would love to address each of your comments and questions.
>
> # Response to Weaknesses:
> > The core issue I have with this paper is that there's very little exploration of the proposed algorithm beyond claims of superior performance. The core concept is simple (not a bad thing), but there's no real consideration given to what makes it good or useful. Seeing as the core contribution of this work is very simple, it should be straightforward to do deeper analysis and benchmarking, but this paper doesn't do much of that, and largely just concludes "it works."
>
> > As a result, I can't help but feel this paper is just sort of insubstantial and doesn't contribute that much to the field. It proposes an idea that is novel, but straightforward as an extension of past work, and doesn't do anything with the idea beyond show that it outperforms the baseline. If accepted as is, I'm not sure how much this will actually move the topic of exploration forward.
>
> > As such, I'm inclined to recommend rejection, but would like to see an expanded version of this work submitted to a future conference, because I do like the core idea of dynamically switching between explore and exploit policies for off-policy RL.
>
> **Response to Weakness**:
>
> We appreciate your thoughtful feedback and positive comments on the novelty and potential of proactively coordinating exploration strategies in off-policy RL. We understand your concerns regarding the depth of analysis and benchmarking, and we aim to clarify the contributions and significance of our work.
>
> The core contribution of this work includes introducing a novel method that proactively coordinates exploration strategies (it switches between two exploration strategies from $A^{SAC} and A^{B}$, rather than switches between explore and exploit policies) and analyzing how this coordination addresses key challenges in combining SAC with curiosity-based exploration. Specifically, we provide insights into the interaction between intrinsic rewards and stochastic policies, demonstrating that our approach mitigates these challenges. While the concept is simple, its simplicity is a strength, making the method easily adaptable to a range of existing curiosity-based exploration methods.
>
> In our experiments, beyond merely demonstrating superior performance, we also highlight:
> * **Adaptability**: We apply KEA in RND and NovelD, showcasing its adaptability and ability to improve performance across different exploration paradigms.
> * **Robustness**: Our experiments on **varying UTD Ratios** emphasize KEA's robustness under different shifting dynamics between exploration strategies (as detailed in **Response to Q3**).
>
> During the rebuttal, to provide a deeper understanding of when and why the proposed method is effective, we have conducted new experiments to analyze the sensitivity of KEA to different switching thresholds ($\sigma$), as described in **Response to Q7** and **Response to Q8**. These experiments further illustrate the structured and adaptable behavior of KEA.
>
> Furthermore, we are exploring the application of KEA to other RL algorithms to assess its generalizability. While further time is needed to complete these experiments, we will provide updated results as soon as they are available before the rebuttal ends.

---

> ### Author Response · Authors · 2024-11-26
> **Response to Reviewer tbHf (2/3)**
>
> # Response to Questions:
> > Q1: In figure 3, the SAC line seems to be flat? I am surprised this task cannot be learned at all by SAC, and that the best method only reaches a ~60% success rate. Perhaps the degree of sparsity is not clear to me from the task description. Perhaps that sparsity could be elaborated on somehow?
>
> **Response to Q1**:
>
> Thank you for the question.
>
> The 2D Navigation task takes place on a 41x41 grid with sparse rewards. The minimum steps to reach the goal range from 18 (closest start) to 55 (farthest start). Episodes terminate if the agent hits boundaries, obstacles, or reaches the goal. A key challenge is a narrow 4-cell-wide aisle linking the left and right regions, making it difficult for the agent to discover and access the goal. This sparsity in rewards and constrained exploration space explains why SAC struggles to learn this task.
>
> We also tested SAC with 1 million environment steps under varying UTD ratios, but it consistently failed to reach the goal, further highlighting the difficulty of this environment for methods lacking effective exploration mechanisms.
>
> In Figure 3, the y-axis represents mean episodic returns during training. While SAC fails to achieve meaningful progress, NovelD and KEA-NovelD both reach a 100% success rate. Notably, KEA-NovelD converges to 100% success earlier, at approximately 160,000 environment steps, compared to NovelD's 250,000 steps.
>
>
> > Q2: I suggest adjusting page spacing so that the critical algorithms 3 and 4 are on the same page, since they are connected and key to this paper's contribution
>
> **Response to Q2**:
>
> Thanks for the suggestion! It is indeed making critical algorithms 3 and 4 more readable. We will make sure the two algorithms appear together in the final formatting of the paper.
>
>
> > Q3: Figure 4 feels like supplemental material to me, since it's basically just validating hyperparameter choices.
>
> **Response to Q3**:
>
> Thank you for the feedback. We apologize for any lack of clarity in our explanation.
>
> KEA primarily addresses exploration inefficiencies arising from the complex interaction between two exploration strategies. While task difficulty and environment design are key factors, exploration efficiency is also influenced by how aggressively the SAC agent and intrinsic reward model are updated.
>
> The UTD ratio affects the evolution of both entropy (which maintains stochastic sampling) and intrinsic rewards, thereby impacting the shifting between the two exploration strategies. When intrinsic rewards impact more than entropy, the agent tends to revisit the same state. Conversely, when entropy has a stronger influence, the agent engages in more random exploration.
>
> Figure 4 highlights KEA's ability to address these challenges by proactively coordinating the two exploration strategies. It demonstrates KEA's robustness under different patterns of shifting between exploration strategies (via varying UTD settings).
>
> We believe the figure provides relevant context for understanding the benefits and trade-offs of our proposed exploration strategy, but we are willing to relegate it to supplementary material if the reviewers disagree.
>
> > Q4: The caption on figure 6 should say what the shaded regions represent.
>
> **Response to Q4**:
>
> Thank you for the feedback! The shaded regions in Figure 6 represent the standard deviation across the evaluation runs. We will update the manuscript to clarify this, along with other suggested revisions.
>
>
> > Q5: Also in figure 6, it looks like the algorithms haven't clearly converged on most of the three tasks. 500k environment steps is not a very large number, perhaps it would be worthwhile to run these experiments for the commonly-used 1 million steps to make sure there's separation at convergence?
>
> **Response to Q5**:
>
> We have started to run DeepMind Control Suite tasks with 1M environment steps but further time is required. We will provide updated results as soon as they are available before the rebuttal ends. Nonetheless, we emphasize that KEA demonstrates significantly faster learning compared to the baselines, particularly under limited environment steps. This faster convergence can be a key advantage in many practically relevant tasks, such as e.g. learning physical systems in robotics problems.

---

> > ### Author Response · Authors · 2024-11-26
> > **Response to Reviewer tbHf (3/3)**
> >
> > # Response to Questions:
> >
> > > Q6: Why is it valuable to vary update to data ratios so much? Is there a reason this is a key hyperparameter for this algorithm? If so that should be explained in the paper. It's not clear how this relates to explore-exploit tradeoffs
> >
> > **Response to Q6**:
> >
> > As mentioned in **Response to Q3**, the Update-to-Data (UTD) ratio plays a critical role in influencing two key factors: entropy (stochastic sampling) and intrinsic rewards (curiosity-based exploration). This directly impacts the shifting dynamics between the two exploration strategies.
> >
> > We acknowledge that this relationship was not sufficiently detailed in the paper and will update the manuscript to clarify the importance of the UTD ratio and its role in exploration. Thank you for bringing this to our attention.
> >
> >
> > > Q7: For that matter, why is sigma (which appears to be a key hyperparameter) not varied? It seems like performance should be sensitive to that parameter since it controls algorithm switching (the core contribution)
> >
> > **Response to Q7**:
> >
> > We conducted experiments to evaluate KEA-RND with different switching thresholds ($\sigma$) using the 2D Navigation task. The results are summarized in the table below.
> >
> > Our findings indicate that while varying the switching threshold does affect KEA's performance, all tested configurations consistently outperform RND (0.292 $\pm$ 0.197). This suggests that KEA is robust to threshold variations within a reasonable range.
> >
> > (For reference, the intrinsic rewards are normalized using a running mean and standard deviation, resulting in a standard value of approximately 1.)
> >
> >
> > | Switching threshold ($\sigma$) | Mean Episodic Return  | Standard  |
> > | ------------------------------ | --------------------- | --------- |
> > | 0.50                           | 0.358455              | 0.151244	 |
> > | 0.75                           | 0.348024              | 0.033442	 |
> > | 1.00                           | 0.407033              | 0.055562	 |
> > | 1.25                           | 0.348026              | 0.149555	 |
> > | 1.50                           | 0.333507              | 0.166823	 |
> >
> >
> >
> > > Q8: Further, when and how much does KEA actually switch between the policies? This is the core phenomenon of this paper, so surely there should be some confirmation it's actually switching regularly and in a structured way, no?
> >
> > **Response to Q8**:
> >
> > Thank you for the question. In our experiments, the usage of $A^{SAC}$ typically ranges from 85% to 90%, while $A^{B}$ is utilized approximately 10% to 15% of the time.
> >
> > The table below provides further details on the ratio of $A^{SAC}$ and $A^{B}$ usage under the conditions described in **Response to Q7**. As the switching threshold ($\sigma$) increases, the usage of $A^{B}$ decreases, since it is only applied in states with very high intrinsic rewards and tends to switch back to $A^{SAC}$ quickly.
> >
> >
> > | Switching threshold ($\sigma$) | Ratio of using $A^{SAC}$ | Ratio of using $A^{B}$   |
> > | ------------------------------ | ------------------------ | ------------------------ |
> > | 0.50                           | 0.7619                   | 0.2381                   |
> > | 0.75                           | 0.8128                   | 0.1872                   |
> > | 1.00                           | 0.8628                   | 0.1372                   |
> > | 1.25                           | 0.8916                   | 0.1084                   |
> > | 1.50                           | 0.9199                   | 0.0800                   |

---

> ### Author Response · Authors · 2024-12-03
> **The Results in Response to Q5**
>
> The results of the extended training for the DeepMind Control Suite tasks to evaluate convergence at 1M steps are summarized in the table below.
>
>
> - Walker Run Sparse
>
> | Method      | Mean    | STD     |
> | ----------- | ------- | ------- |
> | NovelD      | 577.38  | 183.80  |
> | KEA-NovelD  | 846.66  |  98.86  |
>
> - Cheetah Run Sparse
>
> | Method      | Mean    | STD     |
> | ----------- | ------- | ------- |
> | NovelD      | 639.26  | 377.37  |
> | KEA-NovelD  | 870.20  | 162.94  |
>
> - Reacher Hard Sparse
>
> | Method      | Mean    | STD     |
> | ----------- | ------- | ------- |
> | NovelD      | 826.73  | 136.96  |
> | KEA-NovelD  | 872.36  |  55.87  |

---

### Author Response · Authors · 2024-12-02
**Response Summary**

We sincerely thank all the reviewers for their constructive and valuable feedback on our manuscript.

We have made every effort to provide comprehensive and detailed responses to address each of the reviewers' concerns. Furthermore, we have carefully revised the manuscript in accordance with their suggestions, with all changes clearly highlighted in red.

The summary of rebuttal responses and updates made includes:
- **Analysis of Switching Mechanism**.
    - Added new experiments to study the sensitivity of KEA to switching thresholds, showing KEA is robust to threshold variations within a reasonable range. (tbHf, kGRm, and vtHu)
    - Added new experiments to compare with an alternative switching mechanism proposed by reviewer vtHu, showing KEA's efficiency outperforms this alternative approach. (vtHu)

- **Analysis of KEA's Applicability**.
    - Added new experiments to explore KEA's generalization beyond SAC. As the primary exploration, we extend KEA to DQN. (kGRm, Ugcq)
- **Extended Training on DeepMind Control Suite**.
    - During the rebuttal, we extended training for the DeepMind Control Suite tasks to evaluate convergence at 1M steps. (tbHf, Ugcq)

- **Conceptual Clarifications**.
    - We explained the effect during varying UTD ratios and highlighted KEA's robustness under different configurations (tbHf)
    - We explained the reason for choosing RND and NovelD as baselines, emphasizing that these represent novelty-based intrinsic reward exploration, which KEA seeks to improve. (kGRm, vtHu)
    - We clarified the limitations of combining SAC with novelty-based exploration and how KEA addresses these inefficiencies. (vtHu, Ugcq)
    - We clarified the difference between curiosity-based and novelty-based exploration and included more related papers to further complement the discussion of addressing exploration challenges. (vtHu)
    - We improved the clarity in Figures 1 and 6, reformatted the performances into tables, and revised the manuscript to ensure precise terminology and remove redundant text to improve presentation and writing. (tbHf, vtHu)

---

### Meta-Review · Area_Chair_EJPx · 2024-12-20

**Metareview:**

This paper proposes Keeping Exploration Alive (KEA), a new exploration strategy for reinforcement learning (RL) that addresses the explore-exploit dilemma, particularly in sparse reward environments. KEA dynamically switches between two policies: an SAC agent with curiosity-driven exploration and a co-behavior agent with a purely stochastic policy.

Strengths
-----------
- **Well-written:** The paper presents the problem and the proposed solution in a clear and easy-to-understand manner.
- **Novel approach:** KEA introduces a unique mechanism for coordinating exploration by dynamically switching between two different exploration strategies based on the novelty of the encountered states.
- **Promising results:** KEA demonstrates improved performance compared to baseline algorithms in various tasks, including 2D navigation and continuous control.

Weaknesses
---------------
- **Limited analysis:** Despite the novelty of the approach, the paper lacks in-depth analysis of KEA's behavior and the factors contributing to its success. There is little exploration of why and how the dynamic switching mechanism works effectively.

- **Lack of strong baselines:** The evaluation primarily focuses on comparing KEA to its components (SAC, RND, NovelD) rather than including state-of-the-art exploration methods. This limits the ability to assess the true effectiveness of the proposed approach.

- **Questions about generalizability:** The paper focuses on SAC and doesn't discuss how KEA might generalize to other off-policy RL algorithms.

- **Concerns about the switching mechanism:** The use of a fixed threshold for switching between policies raises concerns about its adaptability to different environments and tasks with varying reward sparsity and novelty patterns.

KEA presents an interesting idea for balancing exploration and exploitation in RL. However, the paper needs further development to provide a more thorough analysis of the algorithm's behavior, include stronger baselines in the evaluation, and address concerns about generalizability and the switching mechanism.

**Additional Comments On Reviewer Discussion:**

The discussion has been focused on the generalizability of the proposed method and the extensiveness of the empirical analysis. The rebuttal has been generally appreciated by the reviewers, but not enough to fully resolve their concerns.

---

### Decision · Program_Chairs · 2025-01-22

Reject